# Dynamic transcriptional signature and cell fate analysis reveals plasticity of individual neural plate border cells

**Daniela Roellig\*, Johanna Tan-Cabugao, Sevan Esaian, Marianne E Bronner\***

Division of Biology and Biological Engineering, California Institute of Technology, Pasadena, United States

**Abstract** The 'neural plate border' of vertebrate embryos contains precursors of neural crest and placode cells, both defining vertebrate characteristics. How these lineages segregate from neural and epidermal fates has been a matter of debate. We address this by performing a fine-scale quantitative temporal analysis of transcription factor expression in the neural plate border of chick embryos. The results reveal significant overlap of transcription factors characteristic of multiple lineages in individual border cells from gastrula through neurula stages. Cell fate analysis using a Sox2 (neural) enhancer reveals that cells that are initially Sox2+ cells can contribute not only to neural tube but also to neural crest and epidermis. Moreover, modulating levels of Sox2 or Pax7 alters the apportionment of neural tube versus neural crest fates. Our results resolve a long-standing question and suggest that many individual border cells maintain ability to contribute to multiple ectodermal lineages until or beyond neural tube closure.

**\*For correspondence:** roellig@caltech.edu (DR); mbronner@caltech.edu (MEB)

## Introduction

Evolution of vertebrates is intimately linked to the advent of two embryonic cell types: neural crest and ectodermal placode cells. Both contribute to many of the defining characteristics of vertebrates, including a well-defined head with sensory organs and peripheral ganglia. Neural crest cells differentiate into sensory and autonomic ganglia, pigment cells and elements of the craniofacial skeleton, whereas ectodermal placodes give rise to the ear, nose, lens and sensory ganglia of the head.

In all chordate embryos, the ectoderm in the midline of the embryo, called the neural plate, subsequently folds or cavitates to form the neural tube, the future central nervous system (CNS). In vertebrates, neural crest and placode cells originate from a region of ectoderm at the juncture between neural and non-neural ectoderm, the 'neural plate border'. Whereas basal chordate embryos possess a sharp demarcation between presumptive neural and epidermal fates at this border, much less in known about how cell fates as disparate as neural crest, placode and CNS cells become segregated at the neural plate border of vertebrate embryos.

The classical view, favored in many reviews and textbooks, is that the vertebrate neural plate border initially contains intermingled precursors of various fates that by neurula stages become subdivided into separate zones, with placodal precursors found more laterally and neural crest precursors more medially (*Groves and LaBonne, 2014*; *Moody and LaMantia, 2015*; *Patthey and Gunhaga, 2011*; *Saint-Jeannet and Moody, 2014*; *Streit, 2007*). An alternative and equally possible scenario is that the neural plate border region maintains a random mixture of intermingled but fate-restricted precursors. Finally, it is possible that individual neural plate border cells initially are naïve and capable of giving rise to all ectodermal lineages.

In past decades, this question has been addressed by fate mapping studies in which small groups of neural plate border (*Schoenwolf and Sheard, 1990*) or presumptive placodal precursors

(*Streit, 2002*) were labeled using focal injections of the lipophilic dye DiI in gastrula stage chicken embryos. This results in DiI-labeled cells localized to multiple tissues (e.g. epidermis, neural tube, neural crest and placode). These studies confirmed that the neural plate border region contributes to a mixture of cell fates. However, DiI labeling typically labels small numbers (10-50) of cells, thus leaving open the question of whether the neural plate border contains multipotent and/or inter-mixed, fate restricted cells and their relative positions. While single cell lineage tracing has partially addressed this issue in the trunk neural crest just prior to their emigration (*Baggiolini et al., 2015*; *Bronner-Fraser and Fraser, 1988*; *McKinney et al., 2013*), these studies did not examine early neu-ral plate border stages or the relationship between different transcription factors in the border region.

Here, we characterize dynamic changes in neural plate border cells at high resolution and at the single cell level in the anterior neural plate border of the early chick embryo. To this end, we per-formed quantitative analysis of the transcription factors Sox2, Pax7, Msx1/2, Tfap2a and Six1 in neu-ral plate border cells of chick embryos as a function of time. The results show that individual border cells have a high degree of overlap of multiple transcription factors at early gastrula stages. This overlap is maintained within the rising neural folds until the time of neural tube closure. Cell fate analysis further reveals that cells that initially express Sox2 contribute not only to the neural tube but also to neural crest and epidermis and that modulating transcription factor levels impacts the alloca-tion of cells to neural versus neural crest lineages. Taken together, these results suggest that neural plate border cells have the ability to contribute to multiple ectodermal fates and that segregation of individual lineages does not occur until around the time of or after neural tube closure.

## Results

### Transcription factors associated with different fates are colocalized in the majority of neural plate border cells

In all vertebrate embryos, the neural plate border is the source of both neural crest and placodal precursors. The chick embryo develops as a disc-shaped embryo on top of the yolk such that the gastrula-stage embryo (stages 3–4 accordingly to the criteria of Hamburger and Hamilton, HH) is flat with the presumptive neural region surrounding the primitive streak. In cross section, the neural plate is a single layer of cells surrounding the midline of the embryo, with the neural plate border being relatively broad. Here, we define the neural plate border at mid- to late gastrula stages (HH4 to HH4+) by virtue of the expression of Sox2 throughout the neural plate and Tfap2a, which is expressed throughout the presumptive non-neural ectoderm at this stage. At subsequent stages, we use Pax7 to define the extent of the border, since it is one of the earliest markers of the neural plate border and becomes quickly restricted to this region (*Basch et al., 2006*; *Khudyakov and Bronner-Fraser, 2009*; *Otto et al., 2006*). With time, the neural plate border elevates to form the neural folds as the embryo undergoes neurulation until the time of neural tube closure at HH8+ in the cra-nial region. Neural crest emigration from the neural tube then commences at HH9. The schematic diagram in *Figure 1A* depicts the appearance of the whole embryo as well as transverse sections through the neural plate border as a function of time.

In order to examine whether cells at the neural plate border are biased toward a particular fate (neural crest, placode, neural plate, non-neural ectoderm), we performed immunostaining with a variety of transcription factors associated with these lineages at cellular resolution as a function of time. For this purpose, we chose Sox2, Pax7, Msx1/2, Tfap2a and Six1. Sox2 has been used in many studies to define the neural plate territory (*Basch et al., 2006*; *Linker and Stern, 2004*; *Streit et al., 1997*). Pax7 and Msx1/2 are expressed in the neural plate border and later, Pax7 is expressed in the neural crest and dorsal neural tube (*Basch et al., 2006*; *Khudyakov and Bronner-Fraser, 2009*), whereas Six1 is an early placode marker (*Brugmann et al., 2004*) and Tfap2a is in the non-neural ectoderm, as well as the neural plate border and later in the neural crest (*de Crozé et al., 2011*; *Khudyakov and Bronner-Fraser, 2009*; *Luo et al., 2003*).

Surprisingly, our single cell analyses show that numerous cells at the neural plate border express multiple markers associated with different fates from early gastrula stages and continuing to times of neural tube closure. Frequently, we observed individual cells simultaneously expressing markers

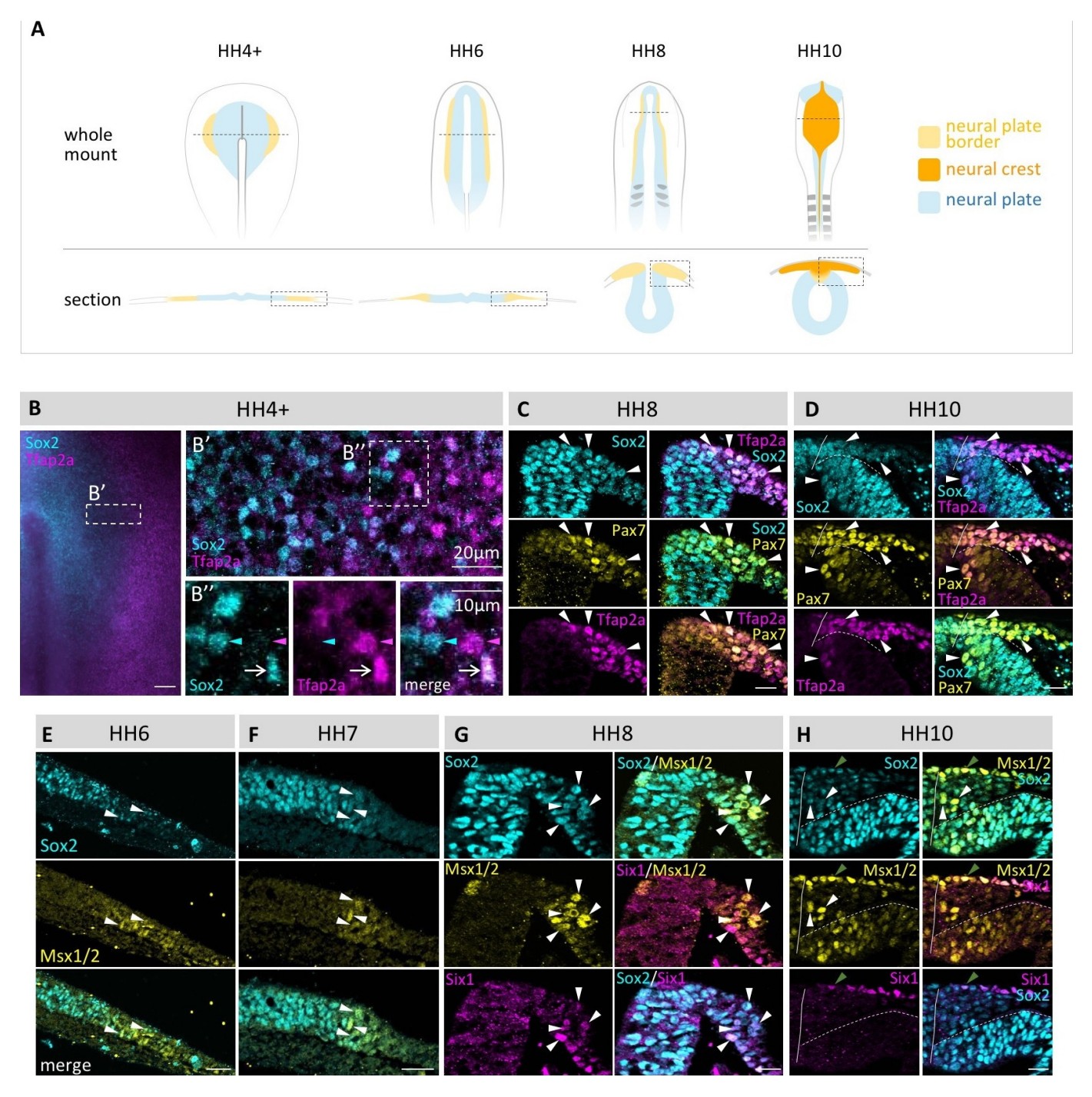

**Figure 1.** Coexpression of multiple transcription factor proteins in individual chick neural plate border (npb) cells as assayed by immunostaining at different developmental stages. (**A**) Schematic diagrams of chicken embryos at different developmental stages from gastrulation to neurulation. Dashed line in whole mount indicates level of section. Dashed boxes indicate area displayed in (**B–H**). (**B–B''**) Sox2 and Tfap2a in whole mount embryo at HH4+. (**B'**) Confocal image of npb (white box in **B**) shows Sox2 (blue) on medial (left) side and Tfap2a (magenta) on lateral (right) side of npb. **B''** zoom of box in **B'** shows an example of a cell coexpressing Sox2 and Tfap2a (arrow) as well as a cell expressing either Sox2 (blue arrowhead) or Tfap2a (magenta arrowhead) only. (**C–D**) Transverse sections of embryos at HH8 to HH10 with co-immunostaining for Sox2 (blue), Tfp2a (magenta) and Pax7 (yellow). Arrowheads indicate examples of cells coexpressing all three markers. (**E–F**) Transverse sections of embryos at HH6 to HH7 with co-immunostaining for Sox2 (blue) and Msx1/2 (yellow). Arrowheads indicate examples of cells coexpressing both markers. (**G–H**) Transverse sections of embryos at HH8 and HH10 with co-immunostaining for Sox2 (blue) and Msx1/2 (yellow) and Six1 (magenta). Arrowheads in (**G**) indicate examples of cells coexpressing all three markers. White arrowheads in (**H**) indicate cells coexpressing Sox2 and Msx1/2 and green arrowhead indicates cell in epidermis co-expressing all

*Figure 1 continued on next page*

*Figure 1 continued*

three markers. All sections are oriented with medial to left and lateral to right side. Scale bars in wholemounts = 100 μm in sections = 20 μm (except 1B'' = 10 μm).

reflective of neural plate, neural crest, placodal and/or epidermal character at multiple stages of development (*Figure 1*).

We observe cells coexpressing Tfap2a and Sox2 as early as HH4+ (*Figure 1B*) and continuing through later stages, e.g. at HH8 and HH10 (*Figure 1C and D*), at which time we also observe concomitant expression of Sox2, Pax7 and Tfap2a in individual neural plate border cells. Similarly, cells co-expressing Msx1/2 and Sox2 can be observed at HH6 to HH10 (*Figure 1E–H*). Co-expression of Msx1/2 with the placodal marker Six1 as well as Six1 plus Sox2 can be observed at HH8 in the dorsal neural folds (*Figure 1G*). At HH10, however, coexpression of Sox2, Msx1/2 and Six1 is only observed in the ectoderm since Six1 is no longer expressed in the neural crest or dorsal neural tube by this stage (*Figure 1H*). Interestingly, in addition to the neural plate territory, expression of Sox2 extends throughout the entire neural plate border region when compared with Msx1/2 and Pax7 at all time points examined (*Figure 1C–H*). Nevertheless, we do also observe some cells that only express a single marker at a time at all stages, though this appears to be a minority population. Altogether, our single cell analysis indicates an extensive overlap in expression of various transcription factors in individual neural plate border cells.

## Quantification of marker coexpression in single cells in the neural plate border

The above results reveal a large degree of overlap of transcription factors associated with diverse lineages in single neural plate border cells. To assess the degree of coexpression of these markers in a quantitative manner, we used Imaris software to quantify how many cells expressed one, two or three markers simultaneously independent of expression intensity. We considered the left and right side of embryos embryos independently and evaluated six neural plate borders (4–5 sections each) of four embryos at HH5, eight neural plate borders (4–7 sections each) of five embryos at HH6, 11 dorsal neural folds (4–6 sections each) of six embryos at HH8 and 7 dorsal neural tubes including delaminated neural crest and Pax7+ cells in epidermis (4–9 sections each) of four embryos at HH9. Based on their predominant marker expression and their antibody compatibility and quality, we evaluated three markers for this analysis: Sox2, Pax7 and Six1 (*Figure 2B–E* and *Figure 2—figure supplement 1*). To unambiguously demonstrate the specificity of the antibodies used, we transfected chicken fibroblast Df1 cells with constructs expressing fluorescently tagged Sox2 or its paralog Sox1 (*Figure 2—figure supplement 2A,B and G*). By immunostaining we find that Sox2 antibody does not recognize its paralog Sox1, even though the construct is transcribed as shown by RT-PCR. A similar experiment was performed for Pax7 and its paralog Pax3 (*Figure 2—figure supplement 2C,D and H*). The Pax7 antibody did not recognize its paralog Pax3. We also show that two different Sox2 antibodies (goat polyclonal Sox2 and rabbit monoclonal Sox2 antibodies) exhibit identical expression patterns (*Figure 2—figure supplement 3*) and according to the manufacturer the rabbit monoclonal Sox2 antibody does not recognize Sox1 or Sox3.

Our coexpression analyses show that ~70% of all cells at the neural plate border express Sox2 and about 75% express Pax7 (*Figure 2F and G*) at HH5. These numbers stay relatively constant over time. However, more than half of these cells (~44%) coexpress Sox2/Pax7 at HH5, which rises significantly to about 54% at HH6 and remains relatively constant until HH9 (~58%) (*Figure 2I*). This suggests an extensive overlap of neural/neural crest precursors.

On the lateral side of the neural plate border, where Six1 is expressed predominantly, we also found cells that coexpress Six1 and Sox2 (*Figure 2—figure supplement 1A–E*), as well as triple expression of Six1/Sox2/Pax7 (*Figure 2B–D* arrowheads). Cells coexpressing Sox2/Six1, but not Pax7 ranged between 0–1%, and cells coexpressing Six1/Pax7 but not Sox2 between 1% (HH6) and 4% (HH9) (*Figure 2—figure supplement 1A–D and F*). However, cells coexpressing all three markers (Sox2, Pax7 and Six1) peaked at 7% at HH8 (*Figure 2J*). Similarly, we see a higher degree of coexpression of Sox2, Pax7 and Tfap2a (*Figure 2—figure supplement 1G*). These data indicate a burst of triple marker coexpression between HH7 to HH8.

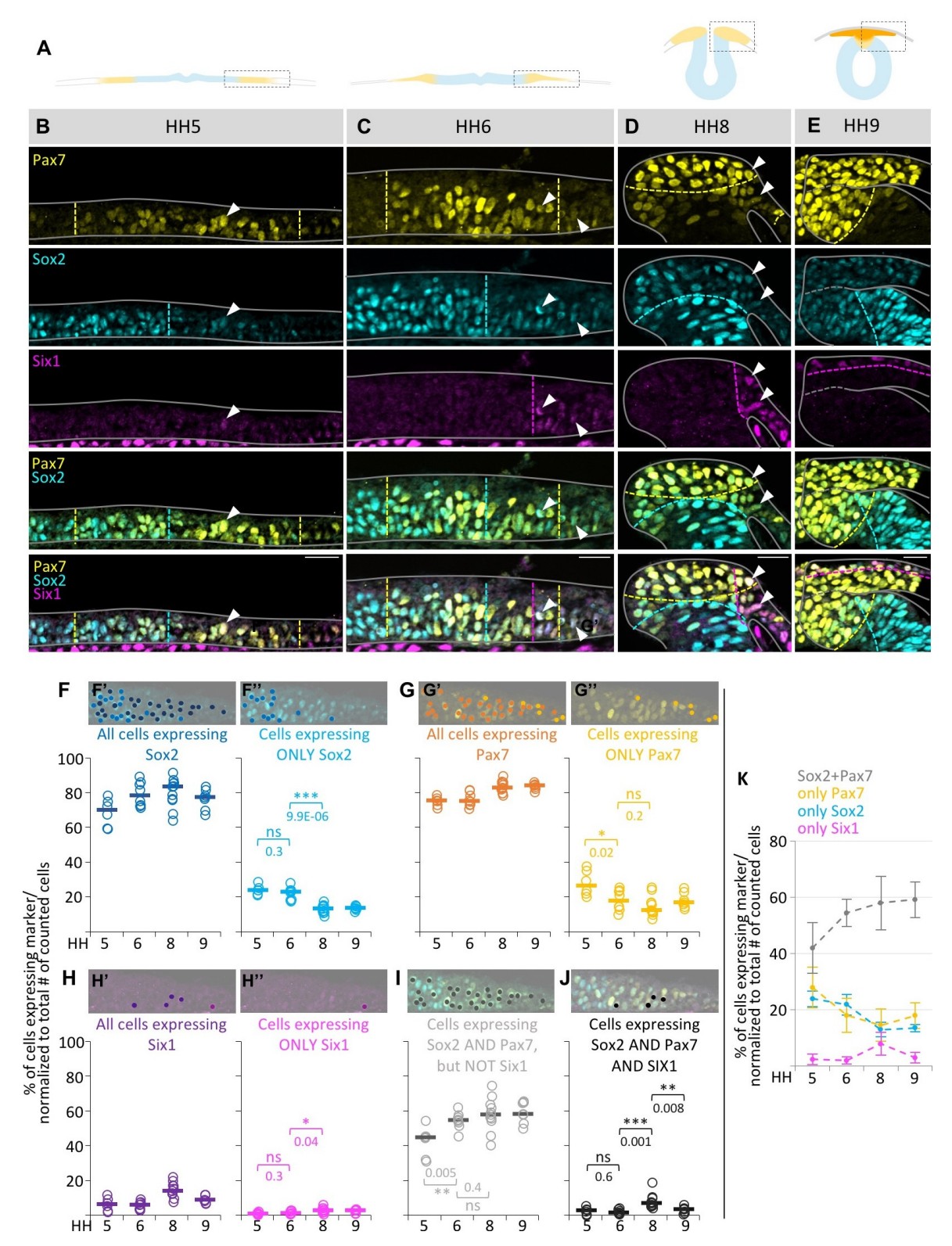

**Figure 2.** Quantification of marker (co-) expression in single cells. (A) Schematic diagrams of sections staged accordingly to those used in (B–E). Box indicates area that was imaged in (B–E). (B–E) Transverse sections immunostained for Pax7 (yellow), Sox2 (blue) and Six1 (magenta) at HH5 (B), HH6 (C), HH8 (D) and HH9 (E). Grey lines outline the embryo borders. Dashed colored lines indicate borders between strong and weak/no expression of corresponding markers. Sections are oriented medial (left) to lateral (right). See also *Figure 2—figure supplement 1*. (F–H) Scatterplots of

*Figure 2 continued on next page*

*Figure 2 continued*

quantification of all cells that express Sox2 (F') , Pax7 (G') and Six1 (H') and the fraction of cells expressing only Sox2 (F''), Pax7 (G'') or Six1 (H'') at HH5 to HH9. (I) Scatterplots representing the fraction of cells that coexpress Sox2 and Pax7, but not Six1 or (J) Sox2, Pax7 and Six1 at HH5 to HH9. See also *Figure 2—source data 1*. On top of scatterplots are sample images of a HH6 section with dots indicating cells expressing the corresponding marker. (K) Combination of medians of F'', G'', H'' and I to illustrate difference of single (Sox2-blue, Pax7-yellow) or coexpressing cells (Sox2/Pax7-grey) at different stages. Scale bars = 20 μm. Asterisks indicate significance as calculated using a Student's t-test with p-values displayed. Error bars indicate standard deviation.

The following source data and figure supplements are available for figure 2:

**Source data 1.** Quantification of marker coexpression in single cells.

**Figure supplement 1.** Coexpression of markers in single neural plate border cells.

**Figure supplement 2.** Specificity of Sox2 and Pax7 antibodies.

**Figure supplement 3.** Different Sox2 antibodies exhibit identical staining patterns.

Although the majority of cells express multiple markers, we do observe neural plate border cells that express only a single marker at all stages. For example, 23–24% of the cells express Sox2 alone at HH5-6 (*Figure 2F''*). Although one might expect that this number would increase with time as an indication that cells are becoming progressively restricted, instead we find that this number significantly decreases to about 13–14% at HH8-9 (*Figure 2F''*). In the case of Pax7, 26% of the cells express Pax7 alone at HH5 and this significantly decreases to 18% and 12% at HH6 and HH8, respectively, before it slightly increases again at HH9% to 17% (*Figure 2G''*), possibly indicating the onset of neural crest migration. Six1 is expressed alone in about 1% and 3% of the cells at HH5/6 and HH8/9, respectively (*Figure 2H''*).

Altogether, this quantitative analysis shows that roughly double the number of cells express two or more markers when compared to cells expressing only one of the three markers at a time (*Figure 2K*). Moreover, the percentage of cells co-expressing different transcription factors increases over time while the number of cells expressing individual markers within the neural plate border diminishes, contrary to what would be expected if the developmental potential of cells was decreasing with time. The finding that multiple markers associated with different fates increases in individual cells over time suggests that neural plate border remains flexible with respect to cell fate choice.

## Spatial distribution of marker expression at the neural plate border

Next we evaluated the spatial distribution of these transcription factors to investigate regional restrictions of cells with different marker co-expression within the neural plate border domain as a function of time. To this end, we analyzed the overall tissue level expression of the transcription factors with Fiji (*Schindelin et al., 2012*) at stages HH5 (3–8 sections/embryo, n = 5 embryos), HH6 (six sections/embryo, n = 6 embryos) and HH8 (6–7 sections/embryo, n = 6 embryos). Since immunostaining reveals relative amounts of fluorescence, we normalized the expression levels of a given factor to a reference area to account for global differences in protein expression levels. Although markers are plotted in the same chart for better comparison of different stages, this is a measure of relative levels of each factor compared with its own reference rather than a comparison between factors. We used Pax7 expression to define the extent of the neural plate border (HH5-6) and later the dorsal neural folds (HH8) (*Figure 3A–F*). We found that the neural plate border narrows over time, such that at HH5 Pax7 expression at the neural plate border is wider than at later stages of development (*Figure 3G–I*). Similarly, the intensity of Pax7 expression increases over time between HH5 to HH6 (*Figure 3G–H*). Within the dorsal neural fold, there is no significant increase of Pax7 expression between HH8 (5–9 sections/embryo, n = 7 embryos) and HH9 (3–6 sections/embryo, n = 5 embryos), but Pax7 expression increases significantly in the delaminating neural crest, which could indicate a time point of fate determination at this stage (*Figure 3J*).

Next, we assessed how Sox2 expression changes over time. Sox2 intensity profile remains relatively constant over time between HH5 and HH8 (*Figure 3G–I*). However, between HH8 and HH9,

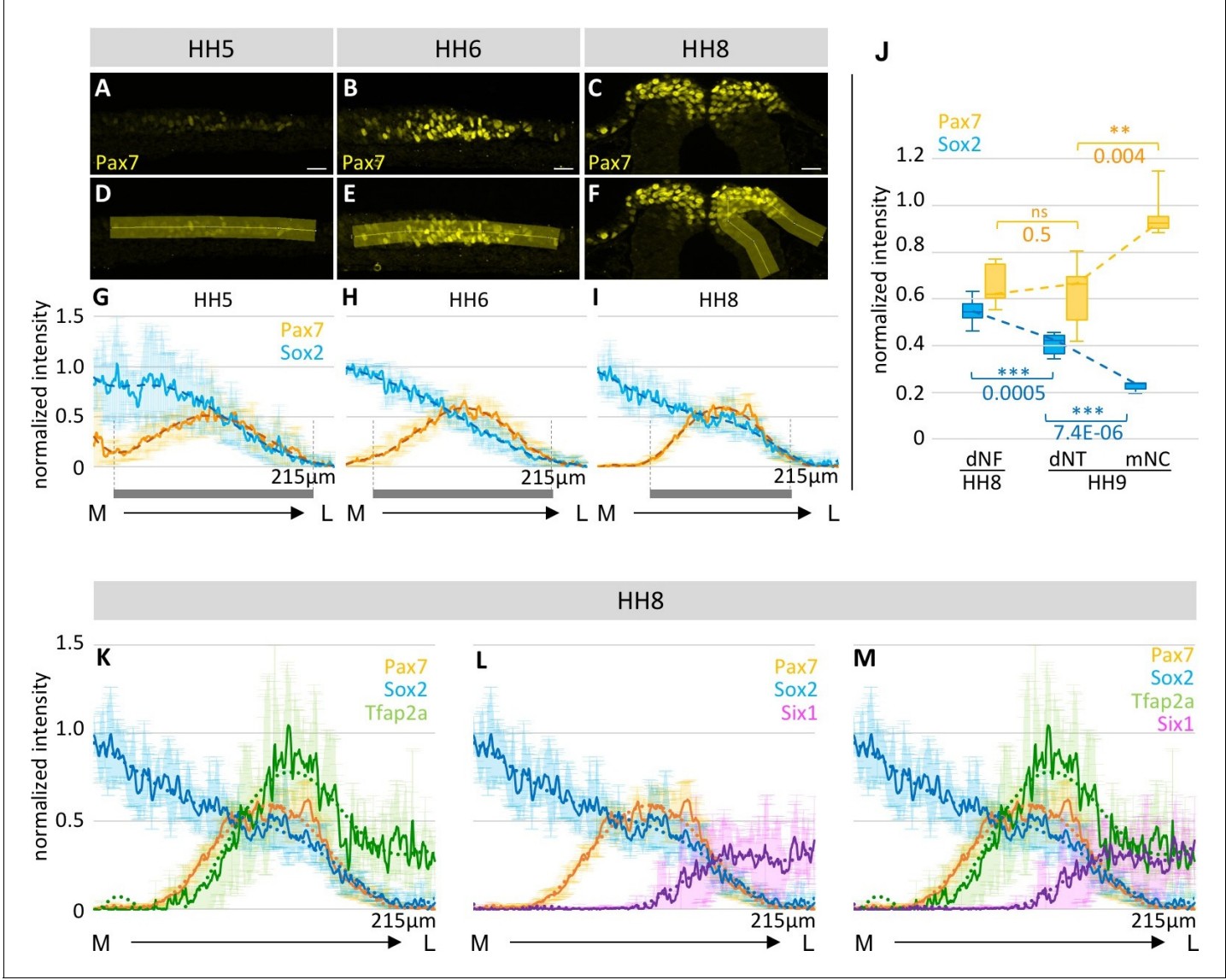

**Figure 3.** Spatial distribution of transcription factor expression at the neural plate border and dorsal neural fold/tube. (**A–C**) Images of Pax7 immunostaining (yellow) in representative embryos used to measure intensity profiles across the neural plate border at HH5 (**A**), HH6 (**B**) and the dorsal neural fold at HH8 (**C**). Scale bar = 20 μm. (**D–F**) A line (yellow), as illustrated in this example, was drawn across the neural plate border/neural fold to measure intensity. (**G–I**) Intensity profiles of Sox2 (blue) and Pax7 (yellow) protein expression across the neural plate border from medial to lateral. Grey bars indicate extent of Pax7 expression at the neural plate border/dorsal neural fold. See also *Figure 3—source data 1–3*. (**J**) Box plot of intensity of Sox2 (blue) and Pax7 (yellow) protein expression in dorsal neural fold (dNF) at HH8 compared to dorsal neural tube (dNT) and migrating neural crest (mNC) at HH9. Asterisks indicate significance as calculated using a Student's t-test with p-values displayed. Error bars indicate standard deviation. See also *Figure 3—source data 4*. (**K–L**) Line plots of relative intensity of certain markers in combination with Sox2 (blue) and Pax7 (yellow) at HH8: (**K**) Tfap2a (green) (2–8 sections per embryo, n = 4 embryos) (**L**) Six1 (magenta) (5–8 sections per embryo, n = 5 embryos). (**M**) Combination of all markers Tfapa2 (green), Six1 (magenta) Sox2 (blue) and Pax7 (yellow). See also *Figure 3—figure supplement 1* and *Figure 3—source data 5* and **6**. (**A–D**) Dark line indicates average across all embryos, with standard deviation indicated by shaded region. Dotted line is polynomial trend line (order 6). M-medial, L-lateral. Intensities are displayed as grey values as measured by Fiji.

The following source data and figure supplement are available for figure 3:

**Source data 1.** Spatial distribution of transcription factor expression at the neural plate border at HH5.

**Source data 2.** Spatial distribution of transcription factor expression at the neural plate border at HH6.

**Source data 3.** Spatial distribution of transcription factor expression at dorsal neural fold at HH8.

*Figure 3 continued on next page*

*Figure 3 continued*

**Source data 4.** Comparison of Sox2 and Pax7 protein expression at HH8 vs HH9.

**Source data 5.** Spatial distribution of Tfap2a expression at the neural plate border at HH8.

**Source data 6.** Spatial distribution of Six1 expression at the neural plate border at HH8.

**Figure supplement 1.** Spatial distribution of expression of different markers across dorsal neural fold at HH8.

Sox2 expression in the dorsal neural fold changes significantly and then drops again in neural crest cells that have delaminated from the neural tube (*Figure 3J*). At HH10 Sox2 expression is maintained in the dorsal neural tube even as neural crest cells begin to migrate away (*Figure 4A*). Interestingly, we also find Sox2/Six1 double and Sox2/Six1/Pax7 triple positive cells in the epidermis (*Figure 4A*). When immunostaining HH11 and HH12 embryos for Sox2 expression, we find that Pax7 positive cells in the dorsal neural tube still express Sox2 even though most of the neural crest cells have already migrated away from the dorsal region (*Figure 4B and C*). Low levels of Sox2 can also be observed in early and late migrating neural crest cells(*Figure 4B and C*).

In comparing the spatial distribution of transcription factors at the neural plate border from medial to lateral, Pax7 is most medial, followed by Tfap2a, then Msx1/2 with Six1 being lateral-most (*Figure 3K–M* and *Figure 3—figure supplement 1*). However, there is an area on the lateral side of the neural plate border where the expression of all these transcription factors overlaps, suggesting that there are cells that potentially express all five markers simultaneously.

Taken together, these data suggest that there are dynamic changes in levels of expression of markers associated with different fates in the neural plate border/neural fold cells at the tissue level, suggesting that the broad developmental potential of neural plate border cells might be due to changes in levels of key transcription factors.

## Evaluating fate of prospective neural plate border cells using a Sox2 neural enhancer

Given the extensive overlap of marker expression in individual neural plate border cells, we asked whether cells that express Sox2 have the ability to form several ectodermal derivatives or only contribute to the CNS. To address this, we sought to examine the subsequent fate of cells that initially express Sox2 at gastrula through neurula stages. For this purpose, we used the well characterized N1 and N2 Sox2 enhancers to drive reporter expression in a manner that recapitulates endogenous *Sox2* gene expression only in the neural plate and neural tube (*Uchikawa et al., 2003*). To build a Sox2-reporter that marks the entire neural plate/tube, we combined the N1 and N2 Sox2 enhancers into a construct that drives H2B-eGFP. eGFP protein is highly stable and has a half-life of approximately 26 hr in mammalian cells (*Corish and Tyler-Smith, 1999*); addition of a H2B nuclear localization signal has been reported to further stabilize the fluorescent label (*Foudi et al., 2009*; *Kanda et al., 1998*), making it an advantageous tool to trace cell fates. Because these enhancers do not drive expression in the neural crest (*Uchikawa et al., 2003*), we were able to follow the contributions of cells that initially express *Sox2* to embryonic tissues (e.g. neural tube, neural crest, and/or ectoderm) at later times.

After introduction of the N1N2-H2BeGFP Sox2-reporter into HH3-4 embryos, the first eGFP protein signal is visible 3 hr post electroporation. The reporter is clearly expressed in the neural plate at HH5 and later in the neural fold/tube including the dorsal portion (*Figure 5A*) as expected and previously described (*Uchikawa et al., 2003*). In addition, we find that enhancer-driven H2BeGFP expression remains in the migrating neural crest as well as a few epidermal cells at HH12 and that signal is even detectable as late as HH14 (*Figure 5B*).

To follow Sox2 expressing cells as function of time in live embryos, we performed time-lapse imaging to visualize the movement of cells expressing the Sox2 reporter. The results reveal streams of N1N2-H2BeGFP expressing neural crest cells emerging from the dorsal neural tube (*Video 1*) and migrating into the facial region. These results confirm that neural plate cells that initially express

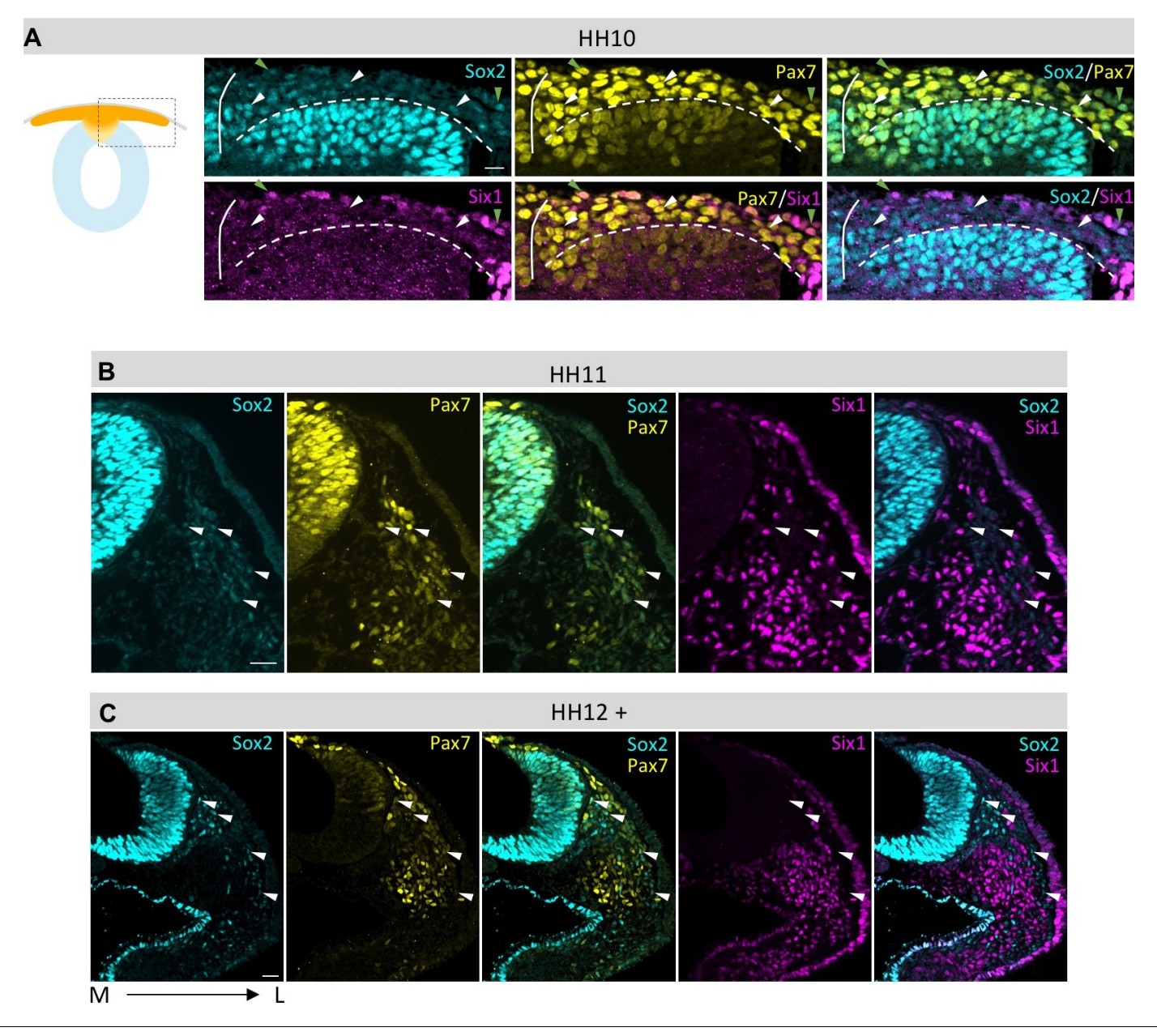

**Figure 4.** Comparison of Sox2, Pax7 and Six1 protein expression in dorsal neural tube and migrating neural crest. (**A–C**) Sox2 protein expression compared to Pax7 and Six1. Low Sox2 levels are detectable in early (**A**, HH10) and late (**B**, HH11; **C**, HH12) migrating neural crest. All sections are oriented medial (left) to lateral (right). White arrowheads indicate delaminated neural crest cells that coexpress Pax7 and Sox2, but not Six1. Green arrowheads (in **A**) indicate epidermal cells that coexpress Pax7 and Sox2 and Six1. White line in (**A**) demarcates the embryonic midline and dashed line represents the border between neural tube and migrating neural crest. Scale bars = 20 μm.

Sox2 contribute to large numbers of neural crest cells that leave the neural tube and populate the periphery.

The above experiments were performed by introducing the enhancer construct into the early gastrula stage embryo. We next asked whether the ability of H2B-eGFP labeled cells to contribute to the neural crest decreases with time. To this end, we electroporated the N1N2 Sox2-H2B-eGFP reporter at progressively later stages. We targeted the closing neural tube at HH7 (2ss) to HH8 (4ss) in ovo, reflecting stages just before and after neural tube closure, respectively. Interestingly, we

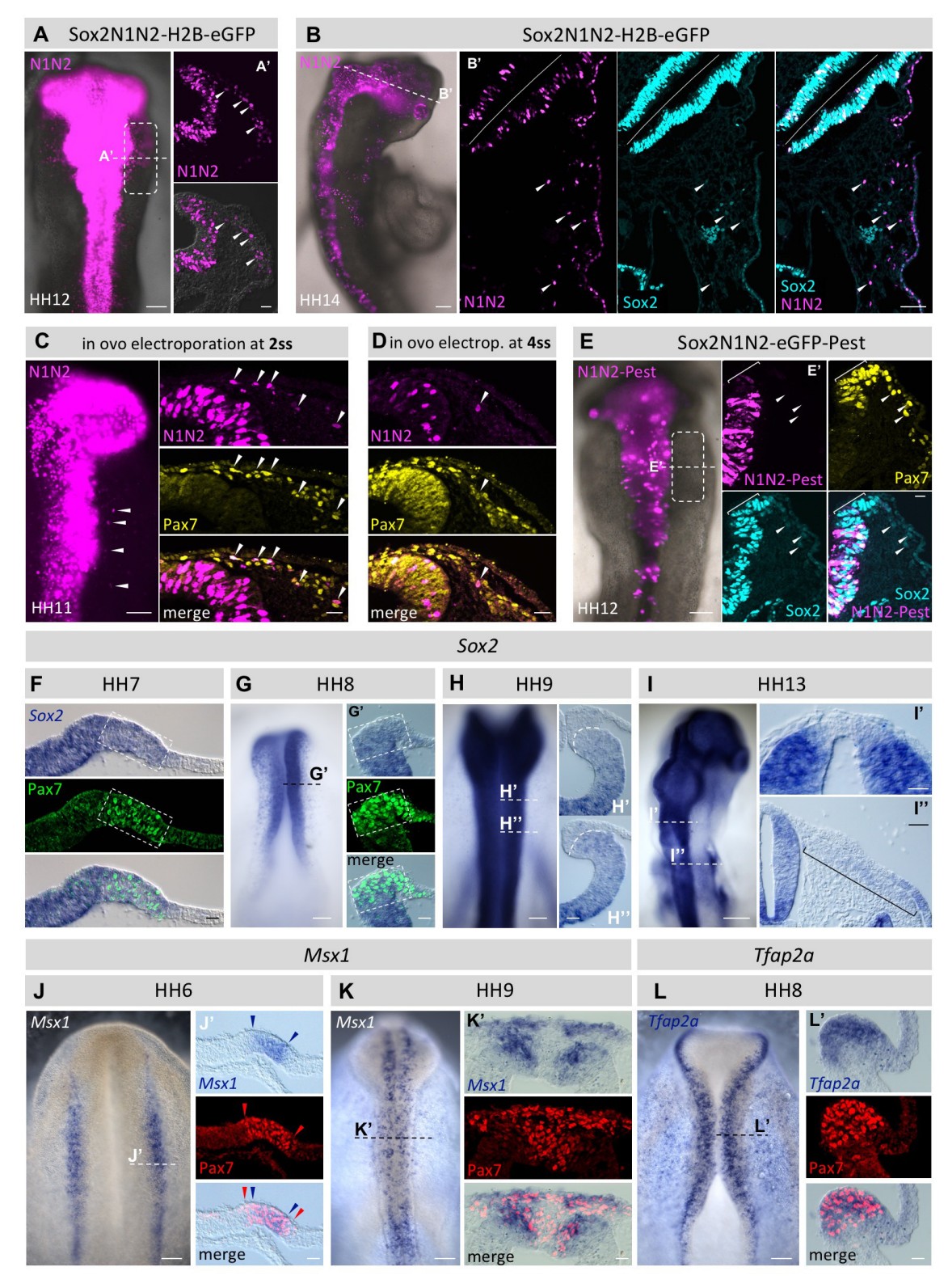

**Figure 5.** Using the Sox2-reporter to follow the fate of neural plate border cells. (**A**) Sox2N1N2-H2B-eGFP construct was electroporated in HH4 chicken embryos. At HH12, eGFP reporter expression is visible not only in the neural tube, but also in migrating neural crest cells (box). Dashed line indicates level of transversal section in (**A'**). Arrowheads indicate N1N2-reporter positive migrating neural crest cells. (**B**) N1N2-reporter expression is maintained in HH14 cranial crest. Dashed line indicates level of section (**B'**). Arrowheads indicate cells positive for N1N2-reporter and endogenous Sox2 protein.
*Figure 5 continued on next page*

*Figure 5 continued*

(C, D) In ovo electroporation of Sox2-N1N2-H2B-GFP 2ss (C) and 4ss (D). Note that the number of electroporated cells decreases at progressively later stages. Arrowheads indicate migrating neural crest cells positive for reporter expression. (E) N1N2-eGFP-PEST was electroprated into HH4 embryos. At HH12, N1N2-eGFP-PEST-reporter is expressed in neural tube, but barely visible in its dorsal most portion (bracket in E'), although endogenous Sox2 protein expression is maintained as visualized by immunostaining (blue). Arrowheads indicate migrating neural crest that express low levels of endogenous Sox2 protein but lack the N1N2-destabilized reporter expression. (F–I) *Sox2* mRNA expression. Dashed lines indicate levels of transverse sections (G', H', H'', I', I''). Transverse sections of (F) HH7 and (G) HH8 embryos showing strong *Sox2* mRNA expression in the dorsal neural fold that overlaps with Pax7 protein expression (box). (H) At HH9 *Sox2* mRNA is barely detectable in the dorsal neural tube and premigratory crest (H', H''). Dashed line in H' and H'' demarcates boundary of dorsal neural tube and pre-migratory neural crest. (I) At HH13 *Sox2* mRNA is barely detectable in the dorsal neural tube (I'), but seen at very low levels in the rhombomere 4 neural crest stream (bracket in I''). (J, K) mRNA expression of *Msx1* in wholemounts. Dashed lines indicate levels of sections (J' and K'). (J') HH7 embryo showing strong *Msx1* mRNA expression (blue arrowheads) in the dorsal neural fold that is nested within the Pax7 protein domain (red arrowheads). (K) HH9 embryo with overlapping *Msx1* mRNA expression and Pax7 protein expression in the dorsal neural tube and the delaminated neural crest. (L) mRNA expression of *Tfap2a* in wholemount at HH8 with dashed line indicating approximate level of section (L'). Scale bars on whole mounts = 100 µm; on sections = 20 µm, except (2B' = 100 µm; F'' = 40 µm).

observe migrating neural crest cells that expressed the eGFP reporter after electroportion at both stages (*Figure 5C,D*), indicating that cells continue to produce eGFP protein under control of these neural enhancers until time points just before neural crest migration initiates. These results suggest that neural plate border/neural fold cells that express Sox2 are able to give rise to not only CNS but also neural crest cells at the time of lineage labeling.

## Sox2 enhancer driving destabilized GFP is not expressed in migrating neural crest cells

Next we assessed whether the strong Sox2-reporter expression is produced in the migrating neural crest cells themselves or is maintained from an earlier time point. To test this, we electroporated a destabilized version of eGFP by adding a PEST domain (N1N2-eGFP-Pest). The Pest domain has been reported to reduce the half-life of GFP to 9.8 hr in mammalian cells (*Corish and Tyler-Smith, 1999*). After electroporation of the destabilized Sox2-reporter, we observe only a very low level of enhancer driven eGFP expression in the dorsal neural tube (*Figure 5E*). In contrast, endogenous Sox2 expression is readily detectable by immunostaining (*Figure 5E'*, bracket). Consistent with this, the N1N2-reporter is not detectable in migrating neural crest cells, whereas low levels of Sox2 protein expression remain (*Figure 5E'*, arrowheads). This indicates that Sox2 protein expression in neural crest cells perdures from transcription and translation occurring prior to migration rather than being newly produced.

We next evaluated how *Sox2* mRNA expression corresponds to Sox2 protein expression in the neural plate border, neural tube and migratory neural crest by performing in situ hybridization (ISH) at different developmental stages. At HH7 and HH8 (*Figure 5F,G*) we observe strong *Sox2* mRNA in the neural plate border and dorsal neural folds and complete overlap with the Pax7 protein expression domain as detected by immunostaining after in situ hybridization. At HH9 *Sox2* mRNA expression in the dorsal neural tube is greatly reduced to almost undetectable levels (*Figure 5H*). This drop of Sox2 mRNA expression between HH8 and HH9 is consistent with the decline of Sox2 protein at that stage (*Figure 3J*) although it remains detectable. At HH13 we no longer see *Sox2* transcripts in the dorsal neural tube (*Figure 5I'*), whereas Sox2 protein is still present in that region likely due to protein stability after *Sox2* mRNA is degraded. *Sox2* mRNA expression is barely detectable in migrating neural crest as it can only be observed at very low levels in the rhombomere 4 stream (pre-otic level) where the neural crest population is very dense (*Figure 5I''*). ISH also reveals overlap of Pax7 and the neural plate border markers *Msx1* and *Tfap2a* at mRNA level. At HH6 we see *Msx1* mRNA expression nested within the Pax7 protein expression domain (*Figure 5J*). At HH9 *Msx1* mRNA expression overlaps with Pax7 protein expression in the dorsal neural tube and the delaminated neural crest. Similarly, Tfap2a mRNA expression overlaps with Pax7 protein expression in the dorsal neural fold at HH8 (*Figure 5L*).

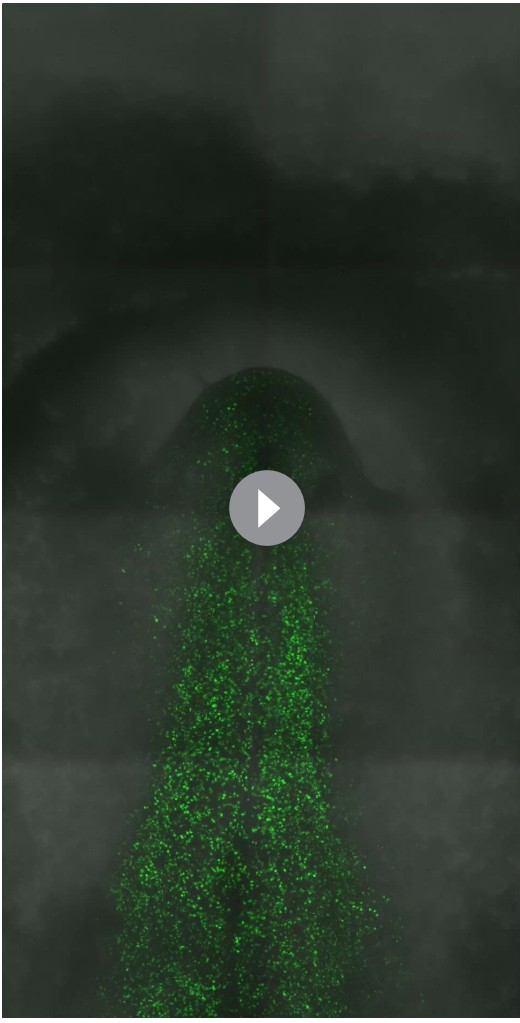

**Video 1.** Using the Sox2-reporter to follow the fate of cells in neural plate border. 4.6 μg/μl Sox2-N1N2-H2B-mEOS3.2 (green fluorescent) was electroporated in HH4- embryos. Maximum projection of 21 z-stacks (24 μm sections, 16 μm interval). 74 timepoints with 15 min time interval (total 18.5 hr). 2 × 3 tiles with 10% overlap. Video runs form stage HH8- to HH11. GFP positive neural crest cells stream away from the neural tube during the last third of the video.

## Manipulation of Sox2 and Pax7 levels affects the balance of neural crest versus neural tube cells

To investigate possible genetic interactions between Sox2 and Pax7, we performed gain- and loss-of-function experiments by electroporating one side of HH4 embryos with a Sox2 overexpression construct pCaggs-cSox2 (n = 40 embryos) with the uninjected side serving as internal control (*Basch et al., 2006*). As a control, we electroporated pCIG plasmid (n = 35 embryos), a GFP-tagged derivative of the pCaggs vector (*Niwa et al., 1991*). The results show that overexpression of Sox2 in the neural plate border results in a significant decrease in Pax7 expression (*Figure 6C,D,E*), in contrast to control embryos (*Figure 6A,B and E*). This seems to be cell autonomous since cells expressing a high level of Sox2 in the dorsal neural fold have greatly reduced Pax7 expression (*Figure 6F*).

In the reciprocal experiment, we tested if Sox2 knock-down altered Pax7 expression. To this end, we electroporated one side of HH4 chicken embryos with a Sox2 translation-blocking morpholino (MO) and the other side with a control morpholino at a concentration of 0.5 mM. This resulted in a small but statistically significant decrease in Sox2 protein levels that in turn caused a statistically significant increase of Pax7 protein expression (5–11 sections/embryo, n = 9) (*Figure 6G,H*). Higher concentrations of Sox2 MO resulted in cell death in the neural tube and were not evaluated further. To control for the specificity of Sox2 MO and to demonstrate that the observed phenotype was not due to off-target binding, we co-electroporated the Sox2 MO with pCaggs-Sox2 (4–7 sections/embryo, n = 16 embryos). This resulted in marked rescue of the loss-of-function phenotype, restoring Pax7 expression (*Figure 6—figure supplement 1*). To further demonstrate morpholino specificity and efficacy, we show that the Sox2 MO causes knock-down of a construct with the Sox2 UTR sequence driving RFP expression but has no effect on an analogous construct with the Sox3 UTR (*Figure 6—figure supplement 2A–D,H*).

To investigate if a decrease in Sox2 expression leads to fate changes at later stages, we again electroporated HH4 embryos with control MO and Sox2 MO on left and right side, respectively, and performed ISH for the neural crest markers *Sox10* (n = 26 embryos) and *FoxD3* (n = 14 embryos). We see that in about half of the embryos *Sox10* and *FoxD3* mRNA expression levels are increased on the Sox2 MO electroporated side when compared to the control MO treated side (*Figure 6I–K*), suggesting a change in fate.

Next, we asked whether changes of Pax7 expression influenced Sox2 expression in the neural plate border. Similar to the experiment above, we overexpressed Pax7 using pCI-Pax7-IRES-H2B-RFP which resulted in a small, but significant decrease in Sox2 protein expression (*Figure 7B–D*)

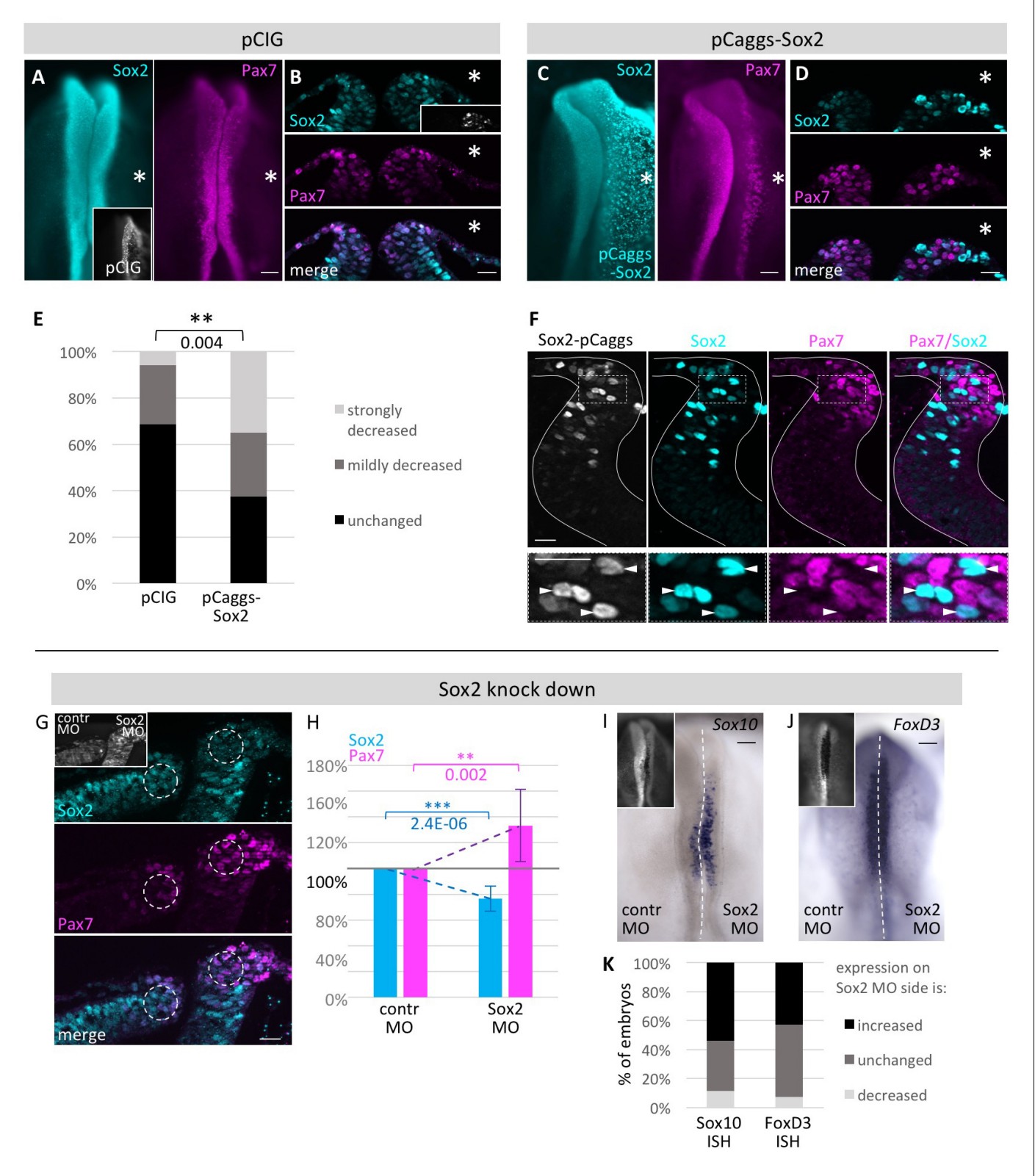

**Figure 6.** Functional analysis of modulating Sox2 levels. HH4 embryos were electroporated with (**A**, **B**) control vector pCIG or (**C**, **D**) pCAGGS-cSox2 on right side of embryo and immunostained for Sox2 (blue) and Pax7 (magenta). Embryos displayed as whole mount (**A**, **C**) or transversal sections (**B**, **D**). (**E**) Quantification of numbers of pCIG or pCaggs-Sox2 treated embryos with a strong or mild loss or unchanged expression of Pax7 versus control side of same embryo. Asterisk indicates significant difference as calculated by contingency table followed by Chi-Square test. Scale bar = 100 μm in whole

*Figure 6 continued on next page*

*Figure 6 continued*

mount and 20 µm in sections. See also *Figure 6—source data 1*. (**F**) Transversal sections of embryos overexpressing Sox2 at HH8. White lines outline neural tube. Box indicates area of enlarged inset. Cells with strong Sox2 (blue) overexpression (arrowheads) in dorsal neural fold show very low levels of Pax7 (magenta). (**G**) Sections of embryo where Sox2 was knocked down with 0.5 mM Fitc-labeled Sox2 MO on right side and control MO on left side of HH4 embryo (inset). Immunostaining for Sox2 (blue) and Pax7 (magenta). Circles in dorsal neural fold indicate area of measurement for (**H**). (**H**) Quantification of Sox2 and Pax7 protein expression in dorsal neural fold upon Sox2 knockdown. Sox2 is reduced by 23.4 ± 9.9% when comparing control to experimental side. This causes increase of Pax7 expression of 33.3 ± 28.0%. Asterisks indicate significance as calculated using a Student's t-test. p-values are indicated in graphs. Error bars signify standard deviation. See also *Figure 6—source data 2*. (**I, J**) Sox2 knock down with 0.5 mM Fitc-labeled Sox2 MO on right side or 0.5 mM Fitc-labeled control MO on left side of HH4 embryo (inset) with subsequent ISH for (**I**) *Sox10* mRNA or for (**J**) *FoxD3* mRNA at HH9. (**K**) Quantification of *Sox10* and *FoxD3* mRNA expression level changes upon Sox2 knockdown. Scale bar = 100 µm. See also *Figure 6—source data 3*.

The following source data and figure supplements are available for figure 6:

**Source data 1.** Quantification/Analysis of Sox2 overexpression experiments.
**Source data 2.** Quantification/Analysis of Sox2 knock down experiments.
**Source data 3.** Quantification of *Sox10* and *FoxD3* mRNA expression upon Sox2 knock down.
**Figure supplement 1.** Loss of Pax7 expression is rescued by exogenous Sox2 protein.
**Figure supplement 2.** Validation of Morpholino knockdown specificity and efficiency.

when compared to the untreated control side of the embryo (3–6 sections/embryo, n = 8 embryos). However, when compared to control embryos electroporated with the empty vector pCI-H2B-RFP (3–6 sections/embryo, n = 7 embryos) (*Figure 7A,C and D*), the difference was not statistically significant. We therefore turned to investigating whether we could detect Sox2 expression level changes on the single cell level. Using Fiji software, we measured the intensity of Sox2 and Pax7 expression in cells with very strong RFP expression in embryos that were injected with Pax7-H2B-RFP on the one side and H2B-RFP on the other side (*Figure 7E*). As expected, cells expressing high levels of RFP also express high levels of Pax7 on the Pax7-H2B-RFP treated side (total of 44 cells of 5 embryos, 2–6 sections evaluated per embryo) but not on the H2B-RFP control side (total of 35 cells of 5 embryos, 4–6 sections evaluated per embryo) (*Figure 7F*). The Pax7-H2B-RFP treated side also had a statistically significant decrease of Sox2 protein expression in these cells when compared to the control side (*Figure 7G*). This indicates a cell autonomous function for Pax7.

In the reciprocal experiment for loss of Pax7, one side of HH4 embryos was electroporated with a Pax7 MO (*Basch et al., 2006*; *Simões-Costa et al., 2012*) and the other side with control morpholino (*Figure 7H*). This resulted in downregulation of Pax7 protein on the Pax7-MO electroporated side after Pax7 immunostaining which in turn caused a significant increase in Sox2 protein expression (5–10 sections/embryo, n = 14 embryos) (*Figure 7I*). When performing ISH for *Sox10* mRNA in MO treated embryos (n = 9 embryos), we see *Sox10* reduction on the Pax7 MO treated side (*Figure 7J–L*), as shown previously at a higher Pax7 MO concentration (*Basch et al., 2006*). To demonstrate specificity, we show that the Pax7 MO effects expression driven by a Pax7 UTR but has little effect on a Pax3 UTR (*Figure 6—figure supplement 2E–I*).

Taken together, these results suggest that there is a reciprocal relationship between levels of Sox2 and Pax7 and that altering these levels affects the balance between prospective neural tube versus neural crest cell fates. This suggests that neural plate border cells can be biased to a different fate by relatively small changes in transcription factor levels.

## Discussion

Transition of a stem cell into a specialized state often involves several cell fate decisions. Transcriptional priming, or the expression of numerous mixed-lineage-affiliated programs, has been described extensively during stem cell fate determination in the hematopoietic system (*Hu et al., 1997*; *Laslo et al., 2006*; *Olsson et al., 2016*). These mixed-lineage states are generally achieved by

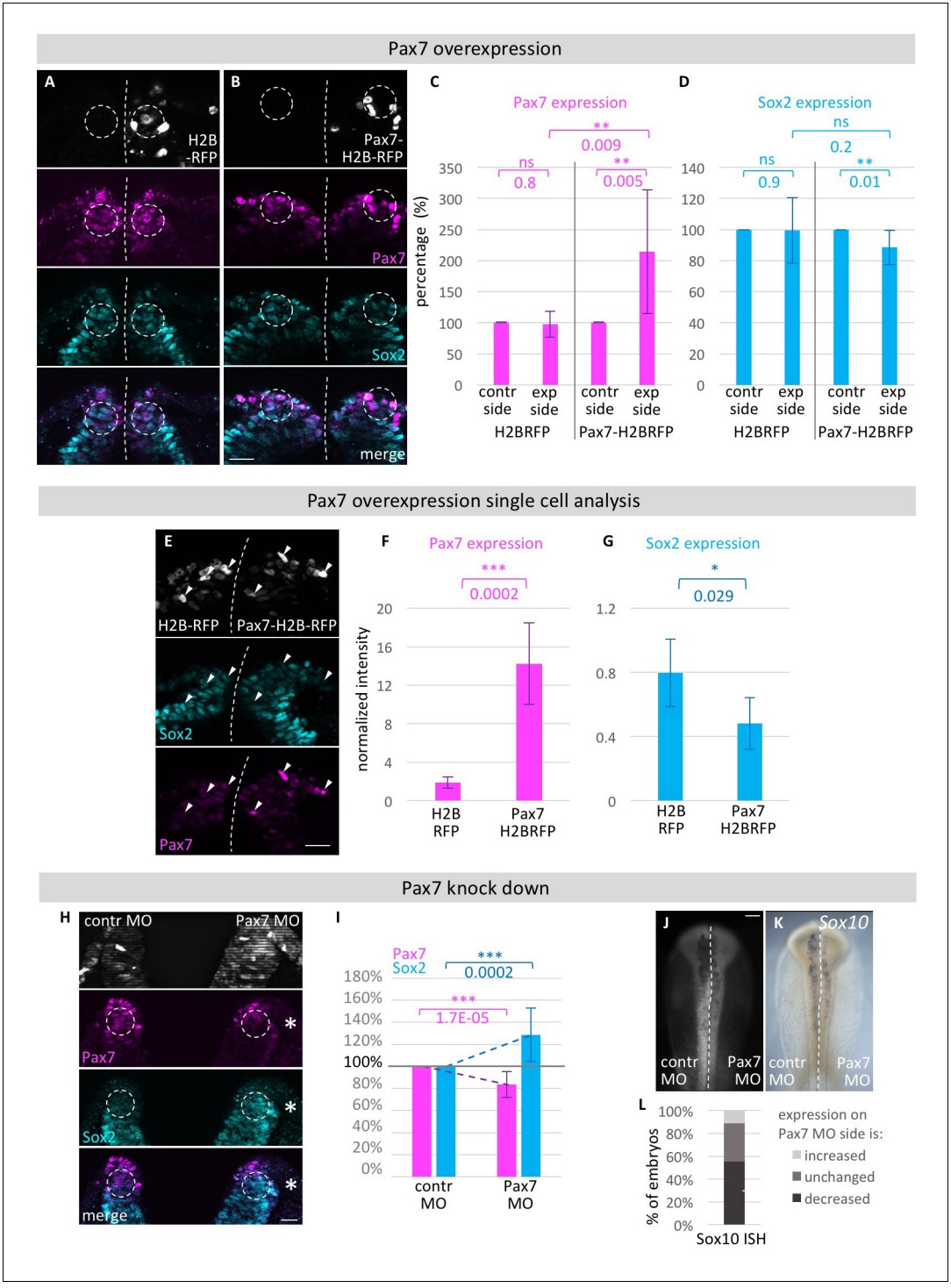

**Figure 7.** Functional analysis of modulating Pax7 levels. HH4 embryos were electroporated with (**A**) control vector pCI-H2B-RFP or (**B**) pCI-Pax7-H2B-RFP on right side of embryo, then immunostained for Pax7 (magenta) and Sox2 (blue) at HH8 and transversely sectioned. Circles in dorsal neural fold indicate area of measurement for C and D. (**C**) Quantification of Pax7 and (**D**) Sox2 protein expression in dorsal neural fold after pCI-H2B-RFP or pCI-Pax7-H2B-RFP overexpression. Pax7 is increased by 115.4 ± 99.3%. Sox2 is reduced by 11.7 ± 11.2% when comparing experimental to control side. Sox2 expression in embryos with Pax7-H2B-RFP treatment on right side versus embryos with H2B-RFP treatment on right side was reduced but not to a level of statistical significance. See also *Figure 7—source data 1*. (**E**) Transversal sections of embryos electroporated with H2B-RFP on left side and Pax7-H2B-RFP on right side for single cell analysis. Quantification of (**F**) Pax7 and (**G**) Sox2 expression in single cells with

*Figure 7 continued on next page*

*Figure 7 continued*

very high levels of RFP expression on H2B-RFP treated side to Pax7-H2B-RFP treated side. See also *Figure 7—source data 2*. (H) Transversal sections of embryo electroporated with 0.5 mM Fitc-labeled Pax7 MO on right side and control MO on left side at HH4. Immunostaining for Pax7 (magenta) and Sox2 (blue). Circles in dorsal neural fold indicate area of measurement for (I). (I) Quantification of Pax7 and Sox2 protein expression in dorsal neural fold upon Pax7 knockdown. Pax7 is reduced by 16.6 ± 11.8% when comparing control to experimental side. This causes an increase of Sox2 expression by 28.6 ± 24.1%. See also *Figure 7—source data 3*. (J) Pax7 knock down with 0.5 mM Fitc-labeled Pax7 MO on left side or 0.5 mM Fitc-labeled control MO on right side of HH4 embryo with (G) subsequent ISH for *Sox10* mRNA at HH9. Scale bar = 100 μm. (L) Quantification of decreased (dark grey), unchanged (medium grey) or increased (light grey) *Sox10* mRNA expression level upon Pax7 knockdown versus control side of same embryo. See also *Figure 7—source data 4*. Asterisks indicate significance as calculated using a Student's t-test. p-values are indicated in graphs. Error bars signify standard deviation.

The following source data is available for figure 7:

**Source data 1.** Quantification/Analysis of Pax7 overexpression experiments.
**Source data 2.** Quantification/Analysis of Pax7 overexpression experiments in single cells.
**Source data 3.** Quantification/Analysis of Pax7 knock down experiments.
**Source data 4.** Quantification of *Sox10* and *FoxD3* mRNA expression upon Pax7 knock down.

expression of alternative lineage determinants like transcription factors and have been proposed to be mandatory steps in cell fate specification. Additionally, single-cell analysis of primary chicken erythroid progenitors revealed a peak of high cell-to-cell variability in gene expression, that precedes an irreversible commitment to differentiation, before being strongly decreased (*Richard et al., 2016*).

Here we show that a similar transcriptional priming appears to occur in the neural plate border, which contains precursors of neural, neural crest, placodal and epidermal cells. By performing an in depth and detailed analysis of differential transcription factor expression at single cell resolution, we have gained a fine-grained view of the signatures of markers indicative of diverse cell fates in this region. Although previous studies have examined gene expression at the neural plate border by in situ hybridization (*Khudyakov and Bronner-Fraser, 2009*; *Streit and Stern, 1999*), the resolution was not sufficient to resolve whether there are characteristic subdomains of cells fated toward neural crest, ectodermal placode, neural or epidermal fates. Our results show that individual neural plate border cells have a high degree of overlap of multiple markers even at early gastrula stages. This overlap is maintained within the rising neural folds until the time of neural tube closure and in some migrating neural crest cells. This suggests that the neural plate border region is not comprised of separate zones containing prospective placodal, neural and neural crest, but rather contains a mixture of intermingled precursors which are not fate-restricted. Consistent with flexibility of cell fates, explant experiments have shown that intermediate neural plate tissue, normally fated to form CNS, can give rise to neural crest cells when exposed to BMP signaling in culture (*Liem et al., 1995*). Thus, we speculate that initially naïve cells are open to many possible fates and retain the ability to contribute to multiple lineages ranging from neural crest, placodal, epidermal and central nervous system fates.

We cannot, however, rule out the possibility that some cells are fate restricted from early stages since a subpopulation was found that only expressed one marker at a time. For example, Pax7 alone was seen in 12–26% of neural plate border cells, Sox2 between 13–24% and Six1 from 1–3% of cells at any particular stage. Although this may indicate commitment to a particular lineage, it is equally possible that stochastic gene expression accounts for this variation in cell to cell protein expression (*Elowitz et al., 2002*; *McAdams and Arkin, 1997*). It is interesting to note that the number of cells expressing single markers did not increase as a function of time, as might be expected with progressive commitment toward a particular fate, but rather dropped. Furthermore, single cells that concomitantly expressed multiple markers were much more prevalent than those expressing a single

factor. Taken together, these results suggest that the majority of neural plate border cells are open to multiple fates.

Classical descriptions of the vertebrate neural plate border at neurula stages draw different domains within the border region, with placodal precursors populating a more lateral domain and neural crest precursors a more medial domain. Consistent with this view, we see Six1 expression primarily in the lateral portions of the neural plate border and Pax7 expression biased toward the medial regions. However, no sharp demarcations exist between precursor populations and we noted extensive intermixing such that even cells expressing a single marker were adjacent to border cells expressing multiple markers. These results are consistent with explant studies performed in mouse embryos. Explants of the prospective neural plate border at a late blastula stage (HH2) were found to express neural, neural crest, epidermal and to a small extent placode markers (*Patthey et al., 2009*). In agreement with these findings, we conclude that the idea of distinct precursor subdomains within the vertebrate neural plate border cannot be substantiated.

In contrast to the chick neural plate border region, neural and epidermal fates in basal invertebrate chordates such as amphioxus are separated by a sharp border. Even urochordates have an invariant cell lineage such that the fate of neural versus epidermal precursors is fixed prior to the gastrula stages (*Conklin, 1905*; *Lemaire, 2009*). However, recent evidence has shown that the caudal neural plate border of *Ciona intestinalis* contains a precursor to bipolar tail neuron (BTNs) (*Stolfi et al., 2015*) that delaminates and migrates along the paraxial mesoderm on either side of the neural tube. These cells share some properties as well as some molecular markers with vertebrate neural crest cells, including co-expression of neural plate border and neural crest markers like *Msx* and *Snail*. Thus, an intriguing possibility is that advent of neural crest and placode cells in the vertebrate lineage resulted in the superposition of a multifated neural plate border region onto a sharp neural/epidermal border in the chordate ancestor and that urochordates may already possess an 'intermediate' neural plate border with partial overlap of marker expression.

To examine the fate of cells that initially express Sox2 in the chick neural plate border, we took advantage of a Sox2-reporter that drives stabilized eGFP expression, enabling long term analysis of the fate of Sox2-positive cells that were electroporated at different times. Although enhancer-mediated expression is turned off as neural crest cells emigrate, the results show that cells initially expressing Sox2 can contribute not only to the central nervous system but also to the neural crest and non-neural ectoderm. This was true for electroporations performed from gastrula stages to the time of neural tube closure. This suggests that Sox2 is present and translated in neural crest precursors until the time they initiate detachment from the dorsal neural tube. These results favor the idea that many individual border cells maintain the ability to contribute to multiple ectodermal fates until the time of neural tube closure. Accordingly, previous single cell lineage analyses in chick (*Bronner-Fraser and Fraser, 1988*; *McKinney et al., 2013*) and mouse (*Baggiolini et al., 2015*) trunk have shown that individual dorsal neural tube cells are multipotent and able to contribute to both neural crest and dorsal neural tube.

Consistent with these lineage studies, our data suggest that the majority of neural plate border cells in vertebrate embryos are open to numerous prospective fates. We further suggest that small changes in transcription factor levels can bias cells toward a particular lineage. For example, we find that subtle modulations of either Sox2 or Pax7 levels can shift the balance of neural versus neural crest cells. This is similar to the well-known Drosophila gap gene scenario where mutually repressive transcription factors regulate each other's relative expression, leading to sharp segmental borders (*Jäckle et al., 1986*; *Jaeger, 2011*). During patterning of the vertebrate neural tube, a transcriptional network of three transcription factors (Pax6, Olig2 and Nkx2.2) defines the final positions of the boundaries of distinct ventral progenitor domains (*Briscoe et al., 2000*, *1999*; *Ericson et al., 1997*; *Novitch et al., 2001*). It was proposed that this three transcription factor circuit encodes a multistable switch that results in stabilizing neural tube patterning (*Balaskas et al., 2012*). Similarly, we propose that the mutual repression of Sox2 and Pax7 can lead to a bistable switch with three solutions (*Figure 8*): an unstable state occurs when the concentration of two cross-repressing factors is similar. However, increasing the concentration of one transcription factor causes a switch from unstable to a stable state where the other one is repressed. According to this model, increase of Sox2 expression above a certain threshold represses expression of Pax7 that in turn leads to neural fate. In the reciprocal situation, high levels of Pax7 induce reduction of Sox2 in the dorsal neural fold and lead to a neural crest fate. In Xenopus a similar model has been proposed for protein cross-

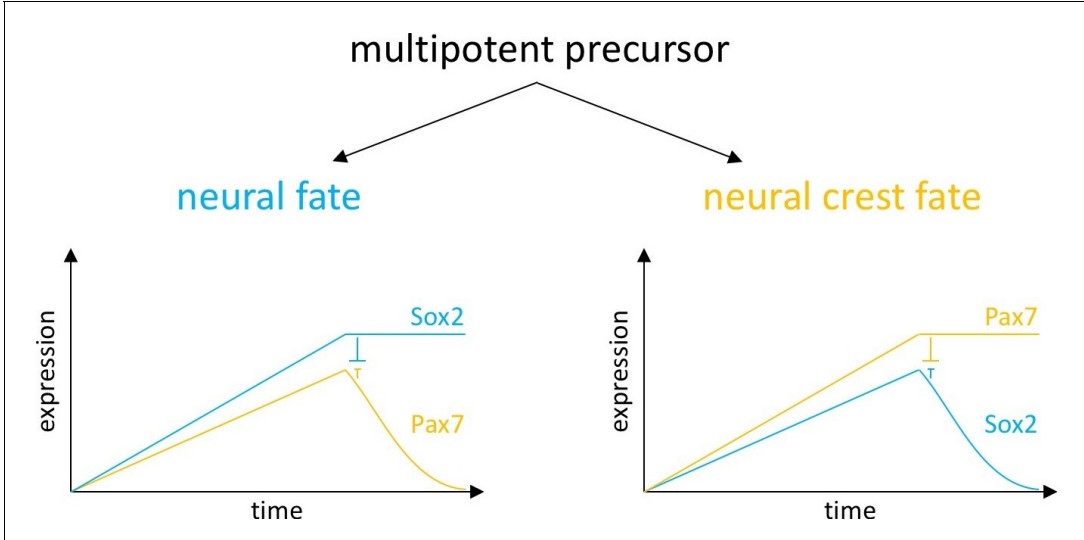

**Figure 8.** Model of fate decisions in the dorsal neural folds. We propose that, within the neural folds, mutual repression can lead to a bistable switch with three possible solutions: an unstable state occurs when the concentration of two cross-repressing factors is similar. At a certain concentration, two different stable states are achieved. When Sox2 expression reaches a certain threshold, it causes reduction of Pax7 and induction of neural fate. When Pax7 concentration increases to a certain level, it causes reduction of Sox2 and induction of neural crest fate.

regulation during cranial ganglion neurogenesis. This work shows that high levels of the placodal genes Eya1 and Six1 promote proliferation of placodal neuronal progenitor by modulating SoxB1 expression, while lower levels promote neuronal differentiation (*Schlosser et al., 2008*).

There are other examples suggesting that levels of transcription factors can affect cell fate at the neural plate border. For instance, Dlx proteins have been shown to shift the position of the zebrafish neural plate border causing changes of cell fates in that region (*Woda et al., 2003*). By inhibiting or ectopically activating endogenous Dlx protein, these authors observed an expansion of the neural plate into the non-neural ectoderm or reduction of neural differentiation, respectively. Similarly, altering BMP levels in frog and chick embryos shifts the position of the neural plate border (*Marchant et al., 1998*; *Streit and Stern, 1999*). Overexpression of Sox2 in the neural folds causes down-regulation of neural crest specifier genes *FoxD3*, *Snail2* and *Sox10* (*Hu et al., 2012*; *Wakamatsu et al., 2004*). Thus, the neural plate border region appears to be highly regulative in ability to respond to transcriptional modifications.

In summary, our data show that the neural plate border contains precursors which have mixed identity until at least the time of neural tube closure. These findings have implications for several fields of biology as correct timing of cell fate decisions and differentiation is not only important for proper embryonic development, but also for stem cell biology, regenerative medicine and cancer cell biology, where new protocols are needed for cell specification and to understand why cells dedifferentiate to return to a progenitor state.

## Materials and methods

### Electroporation

Fertilized chicken eggs were incubated at 37°C to the desired stages. For early stages (HH3-6), embryos were electroporated ex ovo as described (*Sauka-Spengler and Barembaum, 2008*) with five 50 msec pulses of 5.2 V at 100 msec intervals. FITC labeled MO oligonucleotides were obtained from Gene Tools, LLC, Philomath, Oregon. 0.5 to 1.0 mM Sox2 translation blocking MO (5'-ACTAC TTTAGCCTGGAACCAGCCCG-3') or 0.5–1.0 mM Pax7 translation blocking MO (5'-TCCG TGCGGAGCGGGTCACCCCC-3') (*Basch et al., 2006*; *Simões-Costa et al., 2012*) or according amounts of control MO (5'-CCTCTTACCTCAGTTACAATTTATA-3') were electroporated at HH4 and supplemented with carrier DNA (pCIG) at a concentration of 0.5 µg/µl (*Voiculescu et al., 2008*).

Sox2 and Pax7 MOs concentrations above 0.5 mM cause visible detachment of cells from the embryo. Electroporations at HH7-8 (2–4 somite stages) were performed in ovo with five 50 msec pulses of 15V at 100 msec intervals. For over-expression of Pax7 3 µg/µl of pCI-Pax7-IRES-H2B-RFP (gift of Marcos Simoes-Costa) or pCI-H2B-RFP were delivered to one side of HH4 embryos leaving the opposite side as control. For single cell analysis 1.5 µg/µl of pCI-Pax7-IRES-H2B-RFP was electroporated in one side and 1.5 µg/µl of pCI-H2B-RFP to the other side of same embryo. 3 µg/µl of pCaggs-cSox2 (gift of Hisato Kondoh and Fernando Giraldez labs) or pCIG was electroporated on one side of embryo leaving the other side as control. Electroporated embryos were maintained in 35 mm culture dishes with 1 mL of albumen at 37°C in humidified chamber.

## Testing of MO specificity and efficiency

To determine the specificity of the MOs, we first performed a sequence comparison (NCBI blastn suite; https://blast.ncbi.nlm.nih.gov/Blast.cgi) of the Sox2 and Pax7 MO target sequence to the chicken genome and did not find any similarity to closely related family members (e.g. Sox1/Sox3 and Pax3 respectively). To further test specificity and efficiency of MOs, the 5'UTR and coding sequence of the first ten amino acids of the target genes was fused to H2B-RFP (*Simoes-Costa and Bronner, 2016*). The constructs were driven by the strong and ubiquitous pCaggs promoter together with either control or blocking morpholino. Importantly, while doing these experiments, we noticed sequence polymorphism in the Pax7 5'UTR with an extra 'C' within the morpholino recognition site (GGGGGTGACCC'**C**'GCTCCGCACGGA). Therefore, we tested the knock-down ability of the Pax7 MO on both the published UTR sequence and the one with the extra C. In both cases, we observed knock-down though the former was more robust (*Figure 6—figure supplement 2*).

## Cryosectioning

Embryos were incubated in 5% sucrose for 30–60 min at room temperature, in 15% sucrose over night at 4°C and prewarmed gelatin for 2–4 hr at 37°C. Embryos were mounted in silicone molds, frozen in liquid nitrogen and stored at −80°C until sectioning. Thickness of cryosections was 12 µm (immunostained) or 20 mm (ISH). Gelatin was removed by incubating the slides in 1x PBS at 42°C for 10 min.

## Immunostainining

Embryos were fixed in 4% PFA/Phosphate Buffer (PB) for 40–60 min at room temperature. Embryos/ sections were washed 3 × 10 min in PBS/0.5% Triton and blocked for 2 hr in 10% donkey serum. Embryos/sections were incubated in primary antibody solution at 4°C for two or four days, respectively. Primary antibodies used: Six1 (Sigma-Aldrich, St. Louis MO; HPA001893, 1:2000), rabbit anti-RFP (MBL, Woburn, MA; Cat#PM005, 1:1000) and rabbit anti-GFP (Abcam, Cambridge, MA; Cat# ab290, 1:1000) Pax7, Tfap2a and Msx1/2 (Developmental Studies Hybridoma Bank, Iowa City, Iowa; 1:5), goat polyclonal to Sox2 (Santa Cruz Biotech, Santa Cruz, CA; Y-17, 1:1500) for most figures except for *Figure 1C,D* and *Figure 2—figure supplement 3* where a rabbit monoclonal to Sox2 antibody (Abcam, ab92494) was used. Embryos/sections were rinsed twice and washed 4 × 10 min in PBS/0.5%Triton at room temperature on shaker. Secondary antibodies (donkey anti goat IgG Alexafluor488 or 568; donkey anti rabbit IgG Alexafluor488 or 555; donkey anti mouse IgG, Alexa-fluor488/647; goat anti mouse IgG1, Alexafluor647; goat anti mouse IgG2b, Alexafluor 488) were added at 1:1000 in 10% donkey serum/PBS/0.5%Triton and incubated over night at 4°C on shaker. Embryos/sections were rinsed twice and washed 4 × 10 min in PBS/0.5%Triton at room temperature on shaker.

## In situ hybridization

Embryos were fixed in 4% PFA/Phosphate buffered Saline (PBS) at 4°C over night, washed with PBS/ 0.1% Tween20, dehydrated in MeOH, and stored at −20°C. Whole-mount in situ hybridization was performed as described (*Acloque et al., 2008*; *Wilkinson, 1992*). Sox2 dioxigenin-labeled RNA probe was made from plasmid as previously described (*Roellig and Bronner, 2016*).

## Df1 cell culture, RNA extraction and RT-PCR

Df1 chicken embryonic fibroblast cells (ATCC, Manassas, VA; #CRL-12203, Lot number 62712171, Certificate of Analysis with negative mycoplasma testing available at ATCC website) were newly thawed and grown to 60–70% confluence in DMEM (Corning, Tewksbury, MA) supplemented with 10% fetal bovine serum (Gibco, Carlsbad, CA; cat# 26140), 2 mM L-glutamine (Gibco, 25 030–081), and penicillin/streptomycin (1x, Corning, 25–053 CI) at 37°C in 5% $CO_2$. Transfections were performed using the Lipofectamine 3000 Reagent (Invitrogen, Waltham, MA) following manufacturer's instructions for 24 well plates. In short, for each well Opti-MEM Medium was mixed with Lipofectamine 3000 Reagent. A DNA mastermix was prepared by mixing Opti-MEM Medium with DNA and P3000 reagent (2 µl/µg DNA). The two mixes were combined in a 1:1 ratio and incubated for 15 min at room temperature. 50 µl of the DNA-lipid mix were added to 450 µl of supplemented DMEM in each well for a total volume of 500 µl. Triplicates were performed for each condition. Transfection constructs were: pCI-Sox1-GFP (gift of Martin Cheung's lab), pCI-Sox2-H2B-RFP, pCI-Pax3-H2B-GFP (gift of Michael Stark's lab), pCI-Pax7-H2B-RFP (gift of Marcos Simoes-Costa's lab), pCI-H2B-GFP and pCI-H2B-RFP. Df1 cells were incubated in transfection solution overnight then checked for presence of fluorescent cells as a control of successful transfection. Cells were then trypsinized for 5 min in 37°C (0.25% trypsin, Corning, 25–053 CI), centrifuged (10 min at 200 x g) and the supernatant was removed. The cell pellet was resuspended in 100 µl of lysis buffer of the RNAqueous Microkit for RNA Isolation (Ambion, Tustin, CA; Cat# AM1931) and stored at −80°C. RNA was extracted according to manufacturer's instructions and Superscript III First Strand (Invitrogen, Carlsbad, CA; Cat#18080–051) and random hexamer primers were used to obtain cDNA. RT-PCR was performed using the Phusion High-Fidelity PCR Kit (NEB, Ipswich, MA) according to manufacturer's instructions. Primers used were Sox1 (for 5′-AGGAGAATCCCAAGATGCAC-3′, rev 5′-GCCAGCGAGTACTTG TCCTT-3′), Sox2 (for 5′- AGGCTATGGGATGATGCAAG -3′, rev 5′- CTGGATTCCGTCTTGACCAC -3′), Pax3 (for 5′-CGGACGTGGAGAAGAAAATC -3′, rev 5′- GTGCTTCGCCTTCTTGTCTC -3′) and Pax7 (for 5′- ACCACCAACTCCATCTCTGC -3′, rev 5′- GCTTGGCCTGTCTCTACTGG -3′). PCR products were run on a 1% Agarose gel in TAE buffer. A 1 Kb Plus DNA Ladder (ThermoFisher Scientific, Carlsbad, CA) was used for size determination.

## Immunostaining of Df1 cells on cover slips

Transfected Df1 cells were grown to 70–90% confluency on Poly-L-Lysine coated cover slips in 24 well plates. Cells were rinsed in 1 x Phosphate buffered Saline (PBS) and fixed on coverslips by incubating them in 4% Paraformaldehyde in 1x Phosphate Buffer at room temperature for 15 min. Fixed cells were rinsed 2x in PBS and washed in PBS + 0.2% Tween20 for 2 × 10 min, then blocked in 10% donkey serum in PBS + 0.2% Tween20 for 1 hr and incubated in primary antibody overnight at 4°C. Primary antibodies used were mouse anti-Pax7 (Developmental Studies Hybridoma Bank, 1:5), goat anti-Sox2 (Santa Cruz Biotech, Y-17, 1:1500), rabbit anti-RFP (MBL Cat#PM005, 1:1000) and rabbit anti-GFP (Abcam Cat# ab290, 1:1000). Cells were rinsed 2x in PBS and washed 4 × 10 min in PBS + 0.2% Tween20. Secondary antibodies (donkey anti goat IgG Alexafluor488 or 568; donkey anti rabbit IgG Alexafluor488 or 568; donkey anti mouse IgG, Alexafluor647) were added at 1:1000 in 10% donkey serum in 1x PBS + 0.2%Tween20 for 3 hr at room temperature. Cells were rinsed 2x in PBS and washed 4 × 10 min in PBS + 0.2% Tween20. Coverslips were mounted upside down on microscope slides using Fluoromount-G (Southern Biotech, Birmingham, AL) as mounting medium.

## Imaging

Transversal sections were imaged on a Zeiss Imager.M2 with an ApoTome.2 module using HXP 200C illumination. Immunostained sections of embryos of different stages were imaged with 40x Plan-Apochromat, 0.95 Korr (except *Figures 5B* and 20x Plan-Apochromat, N/A 0.8). Whole mount embryos were imaged on same microscope with 5x (EC Plan NeoFluar) or 10x (Plan-Apochromat, N/A 0.45) objective with Axiocam506 mono (fluorescence) and color (DIC/brightfield) cameras (except *Figure 5B*, imaged on Olympus MVX10 with Zeiss MRm camera). *Figure 1B'* was imaged on Zeiss 710 inverted confocal microscope with LD C-Apochromat 40x/1.1 W Korr UV-VIS-IR objective.

## Time lapse imaging

Embryos were electroporated with Sox2N1N2-H2B-mEOS3.2 at HH4⁻ before incubation at 37°C in a humidified chamber for at least 2 hr. A modified paper ring Early Chick culture system (*Chapman et al., 2001*; *Huss et al., 2015*) was used for imaging. Briefly, a thin bed of agar/albumin was poured into two-well Lab-Tek chambered #1 coverglass slides (Thermo Scientific) and allowed to solidify. Embryos were screened under an Olympus MVX10 stereomicroscope for the presence of the fluorescent reporter and normal morphology. Selected embryos were transferred dorsal side down on the agar/albumin bed before returning them to the 37°C incubator for another 2 hr to allow the embryo to settle down into the agar/albumin layer before imaging was initiated. Imaging was performed on a Zeiss LSM710 inverted confocal microscope which had been pre-warmed to 37°C using an electronically controlled environmental chamber (Pecon) and with an EC Plan-Neofluar 10x/ 0.3 objective. Ex 488 (1%)/ Em 488–553; scan speed of 8. Maximum projection of time lapse images was performed in Fiji.

## Quantification

Imaris software (Bitplane, version 8.1.2, 64x) was used to quantify overlapping protein expression in single nuclei. Pax7 expression was used to define the area of interest (extent of neural plate border/ dorsal neural folds) at different stages. By using the spot function, a dot was placed on every nucleus that expressed Pax7. A second and third spot function was created to place dots on every nucleus that expressed Sox2 and Six1, respectively. Spots were placed independent of intensity of markers. Total number of nuclei expressing one marker was displayed by the respective spot function. Over-lapping expression was evaluated by manually counting nuclei with two or three dots. Data were transferred to Microsoft Excel and for each neural plate border/neural fold the average of data collected for the according sections was formed. This was done for each developmental stage. Then, counts for each marker were normalized to the total number of counted cell nuclei per section. This was done for each neural plate border/neural fold for all stages (see *Figure 2—source data 1*). Since the Imaris spot function does not discriminate if nuclei express more than one marker, it was then calculated how many nuclei express one, two or three markers by subtraction. In scatterplots, each circle represents the average of all sections measured for one neural plate border/neural fold and the bar represents the median of all neural plate borders/neural folds per developmental stage.

Fiji software (*Schindelin et al., 2012*) was used to measure intensity of protein expression on Zeiss .czi files. Using the segmented line tool a line of about 215 microns and width of 200 pixels was drawn across the neural plate border from medial to lateral dimension (*Figure 3D–F* and *Figure 3—figure supplement 1*). Intensity was measured as grey values. For *Figure 3J* intensity was measured by placing a fixed-sized oval (1087 square pixel) on dorsal neural fold. Additionally, the intensity of the background and a reference area was measured for each section (reference areas: for Sox2 ventral neural fold, for Pax7 and Msx1/2 highest intensity area in neural plate border/neural fold, Six1 and Tfap2a epidermis on lateral side). In Excel, background was subtracted from marker expression intensity and the intensity of the reference area. Then the ratio between background-adjusted marker intensity and background-adjusted reference intensity was obtained for each position. Negative numbers were changed to zero. The average and standard deviation was calculated for all embryos per developmental stage. The smallest value (Excel Min function) was subtracted from the value in each position to ensure similar levels of graphs on y-axis for better comparability of graphs between different developmental stages. For *Figure 3J*, boxplot was created in Microsoft Excel and displays first to third quartile (box) with median indicated as a line and whiskers that indicate minimum and maximum.

For *Figures 6* and *7*, intensity (grey values; Fiji software) was measured by placing a fixed-sized oval (737 square pixel) on the experimental and control side of the dorsal neural fold. Intensity values of control (contr) sides were set to 100% and experimental values were scaled to the intensity value of corresponding control of each section (value experimental side x 100/ value control side) (*Figure 6—source data 2*; *Figure 7—source data 3*). Values of all sections per embryo were averaged and the standard deviation as calculated in Excel. For single cell analysis (*Figure 7E–G*) a fixed-sized oval (16 square pixel) was placed on cells with very strong RFP expression and intensity measured for the different markers. Intensity was normalized to intensity of reference area (oval of 737 square pixels in the ventral neural tube for Sox2 and the dorsal neural fold for Pax7) for each

marker (experimental intensity/reference intensity). Values of all cells per embryo were averaged and the standard deviation was calculated in Excel. Significances were calculated using a Student's t-test (ns p>0.05; *p≤0.05; **p≤0.01; ***p≤0.001).

## Acknowledgements

We thank Bertrand Bénazéraf for helpful discussions and critical reading of the manuscript. We thank Laura Kerosuo and Erica J Hutchins for advice on cell culture experiments and comments on the manuscript. We are grateful to Fabienne Pituello for providing work space in her lab for revisions. We thank Hisato Kondoh and Fernando Giraldez for the generous gift of the pCaggs-cSox2 construct, Marcos Simoes-Costa for the gift of pCI-Pax7-IRES-H2B-RFP, Martin Cheung for pCaggs-Sox1-GFP and Michael R Stark for the gift of the pCI-Pax3-H2B-GFP plasmid. SE was funded by the CIRM Bridges Training Grant #TB1-01176.

## Additional information

### Competing interests

MEB: Senior editor, *eLife*. The other authors declare that no competing interests exist.

### Funding

| Funder | Grant reference number | Author |
| --- | --- | --- |
| National Institutes of Health | HD037105 | Daniela Roellig<br>Marianne E Bronner |
| California Institute for Regenerative Medicine | CIRM Bridges Training Grant #TB1-01176 | Sevan Esaian |

The funders had no role in study design, data collection and interpretation, or the decision to submit the work for publication.

### Author contributions

DR, Conceptualization, Data curation, Formal analysis, Supervision, Validation, Visualization, Methodology, Writing—original draft, Project administration, Writing—review and editing; JT-C, SE, Investigation, Visualization; MEB, Conceptualization, Resources, Supervision, Funding acquisition, Writing—original draft, Project administration, Writing—review and editing

### Author ORCIDs

Daniela Roellig, http://orcid.org/0000-0002-7558-3592
Marianne E Bronner, http://orcid.org/0000-0003-4274-1862

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
