## [Decision Letter]

Thank you for submitting your article "Dynamic transcriptional signature and cell fate analysis reveals plasticity of individual neural plate border cells" for consideration by *eLife*. Your article has been favorably evaluated by Fiona Watt (Senior Editor) and four reviewers, one of whom, Alejandro Sánchez Alvarado (Reviewer #1), is a member of our Board of Reviewing Editors. The following individual involved in review of your submission has agreed to reveal their identity: Sally Moody (Reviewer #2).

The reviewers have discussed the reviews with one another and the Reviewing Editor has drafted this decision to help you prepare a revised submission.

Summary:

This well-written and elegant body of work represents a thoughtful effort to address the central, yet unresolved issue of whether or not embryonic neural folds contain multipotent progenitors. The study reports on the time course of expression of various transcription factors in the neural border zone during the developmental window when neural plate, neural crest and placode fates should be segregating. The work takes full advantage of available reagents and methods to interrogate the dynamics of key fate determining transcription factors in the development of neural crest cells. By using immunofluorescent detection, the authors uncovered that the majority of cells in the neural border zone (which ultimately assumes neural plate, neural crest and placode fates) express more than one TF indicating that cell fate may remain plastic for longer periods of time than previously anticipated. The findings that cells in the border zone are expressing several TFs simultaneously are novel, and they lead to the important conclusion that these cells are multipotent for longer in development than anticipated from prior studies based on mRNA expression domains (ISH). These findings present important new data for the fields of neural crest and placode development, and have broader implications for all areas, particularly stem cell biology, in which understanding the progression of cell fate restrictions is important.

Essential revisions:

1) Multipotency is a functional definition: to be proven multipotent, a cell has to be shown to give rise to 2 or more different cell types (e.g., single cell lineage tracing, single cell cloning). Early co-expression of two proteins, which happen to be retained later in distinct cell types, is not sufficient to prove multipotency. At most, these variations reflect interesting different regulatory states within a tissue, although potential oscillations in protein content cannot be visualized. The outcome of these diverse regulatory states in terms of cell fate choice remains to be identified. Thus, many conclusions in the manuscript are, unfortunately, over-interpretations and need to be qualified (e.g. at the end of the subsection “Quantification of marker coexpression in single cells in the neural plate border”).

2) Thus far, there is no early (i.e. gastrula stage) specific marker for cell fates arising from the neural plate border. Hence the choice of markers used here is an issue, as there is no marker "predictive of different fates" (subsection “Markers associated with different fates are colocalized in the majority of neural plate border cells”, second paragraph). In chick, Sox2 is expressed in the whole epiblast initially then restricts to the neural plate (Streit et al., 1997). During neurulation, not all neural tube cells maintain Sox2 expression (this study, Figure 4); finally, Sox2 is activated in the placodes by Six1 (Riddiford et al., 2016). Sox2 is thus not a "definitive neural marker" (see aforementioned paragraph), as the authors themselves show in later sections of the manuscript, *Sox2* expression varies according to the stage considered, and many neural crest cell progenitors have expressed Sox2.

Similarly, Pax7 cannot be used as a bona fide neural crest progenitor marker as in many instances in this study (e.g. Introduction, last paragraph; subsection “Quantification of marker coexpression in single cells in the neural plate border”, end of first paragraph), since it is expressed in the dorsal neural tube.

Finally, AP2 is expressed in the early neural plate border (Luo et al., 2003; de Croze et al., 2011), in the premigratory and migrating neural crest, and in the non-neural ectoderm: it cannot be used to specifically identify the non-neural epidermis (Introduction, last paragraph) or the late neural crest (subsection “Quantification of marker coexpression in single cells in the neural plate border”, second paragraph). Thus, the choice of markers is not suitable to support the conclusions of this analysis. Rather, there are interesting and important information on the relative expression levels of these proteins.

3) It is not clear how many embryos were used for some of the key experiments, and equally important how many sections within each embryo were subjected to the analyses reported here. Although numbers are to be found in some of the figure legends, a more explicit identification of these numbers in the main text of the relevant sections of the manuscript would be welcomed.

4) Because of their central importance to support the claims put forward by the authors in this manuscript, it seems appropriate to unambiguously demonstrate the specificity of the antibodies used. In addition, experiments/controls that gauge the affinities of the different antibodies used for quantification should be shown so as to establish whether the resulting quantification data is or is not based by different affinities of the antibodies for their respective epitopes.

5) There is some controversy in the developmental biology field as to the specificity and efficacy of morpholino-based knock-down. Thus, it is important that researchers using these reagents present as much information as possible so their results are not dismissed by skeptics in the field. Some suggestions include: showing by Western blot that the Sox2 and Pax7 MOs significantly decrease protein expression perhaps in a cell line in which the increased amount of MO would not be lethal as it is in the embryo; showing that the MOs will not bind to closely related family members, i.e., simply do a sequence comparison; rewording the legend of Figure 5 to state what percent KD is achieved. Yes, the bar graphs show those data, but the text could be misconstrued. For example, on first read I misinterpreted the figure legend to state that Pax7MOs achieved 81% knock-down. I think it would be beneficial to make the language a little clearer.

6) It has been well described in the *Drosophila* gap gene literature that mutually repressive TFs are expressed in the same cells and regulation of their relative levels eventually leads to sharp segmental boundaries. I see a lot of similarities between that work and what is presented in this manuscript, particularly since Sox2 and Pax7 seem to repress each other. Therefore, reference to this mechanism is appropriate.

7) The number of embryos is rather on the low side; for analytic power in mouse embryos we use 5 embryos per experimental group/genotype as the rule of thumb. Since chick embryos are more readily available, the conclusions would be more convincing if the power were increased by larger sample sizes.

---

## [Author Response]

*Essential revisions:*

*1) Multipotency is a functional definition: to be proven multipotent, a cell has to be shown to give rise to 2 or more different cell types (e.g., single cell lineage tracing, single cell cloning). Early co-expression of two proteins, which happen to be retained later in distinct cell types, is not sufficient to prove multipotency. At most, these variations reflect interesting different regulatory states within a tissue, although potential oscillations in protein content cannot be visualized. The outcome of these diverse regulatory states in terms of cell fate choice remains to be identified. Thus, many conclusions in the manuscript are, unfortunately, over-interpretations and need to be qualified (e.g. at the end of the subsection “Quantification of marker coexpression in single cells in the neural plate border”).*

We apologize for misusing the term multipotency and completely agree with the reviewer that expression of particular transcription factors is not predictive of cell fate. What we meant to say is that many neural plate border cells express diverse and multiple transcription factors. We have now gone through the manuscript carefully to make the language more precise.

*2) Thus far, there is no early (i.e. gastrula stage) specific marker for cell fates arising from the neural plate border. Hence the choice of markers used here is an issue, as there is no marker "predictive of different fates" (subsection “Markers associated with different fates are colocalized in the majority of neural plate border cells”, second paragraph). In chick, Sox2 is expressed in the whole epiblast initially then restricts to the neural plate (Streit et al., 1997). During neurulation, not all neural tube cells maintain Sox2 expression (this study, Figure 4); finally, Sox2 is activated in the placodes by Six1 (Riddiford et al., 2016). Sox2 is thus not a "definitive neural marker" (see aforementioned paragraph), as the authors themselves show in later sections of the manuscript, Sox2 expression varies according to the stage considered, and many neural crest cell progenitors have expressed Sox2.*

*Similarly, Pax7 cannot be used as a bona fide neural crest progenitor marker as in many instances in this study (e.g. Introduction, last paragraph; subsection “Quantification of marker coexpression in single cells in the neural plate border”, end of first paragraph), since it is expressed in the dorsal neural tube.*

*Finally, AP2 is expressed in the early neural plate border (Luo et al., 2003; de Croze et al., 2011), in the premigratory and migrating neural crest, and in the non-neural ectoderm: it cannot be used to specifically identify the non-neural epidermis (Introduction, last paragraph) or the late neural crest (subsection “Quantification of marker coexpression in single cells in the neural plate border”, second paragraph). Thus, the choice of markers is not suitable to support the conclusions of this analysis. Rather, there are interesting and important information on the relative expression levels of these proteins.*

Again, apologies for not being sufficiently clear on this point, and we completely agree that no transcription factor is predictive or characteristic of a particular lineage. For the purpose of telling a story, we were essentially setting up a “straw man” that we then disproved since our results clearly show that the majority of neural plate border cells express multiple transcription factors. Obviously, we were not sufficiently careful with our wording for which we apologize. We have now attempted to correct this and appreciate the reviewer bringing this to our attention.

*3) It is not clear how many embryos were used for some of the key experiments, and equally important how many sections within each embryo were subjected to the analyses reported here. Although numbers are to be found in some of the figure legends, a more explicit identification of these numbers in the main text of the relevant sections of the manuscript would be welcomed.*

Thank you for bringing this omission to our attention. We have now carefully gone through the manuscript to include numbers of embryos, sections examined, etc., and have included more complete methods describing the analysis.

*4) Because of their central importance to support the claims put forward by the authors in this manuscript, it seems appropriate to unambiguously demonstrate the specificity of the antibodies used. In addition, experiments/controls that gauge the affinities of the different antibodies used for quantification should be shown so as to establish whether the resulting quantification data is or is not based by different affinities of the antibodies for their respective epitopes.*

We apologize for not providing this information previously. We have used two different Sox2 antibodies, both of which exhibit identical expression patterns and according to the manufacturer the monoclonal rabbit anti-Sox2 antibody does not recognize Sox1 or Sox3 (see also Figure 2—figure supplement 3). To address this point, we now show that the Pax7 antibody does not recognize its paralog Pax3 and that the Sox2 antibody does not recognize its paralog Sox1. This is now shown in Figure 2—figure supplement 2.

Importantly, we clarify that we are quantifying relative levels of Sox2 to Sox2 at different stages and relative levels of Pax7 to Pax7 at different stages, but not comparing Pax7 protein levels to Sox2 protein levels. Thus, the affinity of the antibody to the protein to which it binds is not relevant in this case. We also now state in the text that although markers are plotted in the same chart for better comparison of different stages, this is a measure of relative levels of each factor compared with its own reference rather than a comparison between factors.

*5) There is some controversy in the developmental biology field as to the specificity and efficacy of morpholino-based knock-down. Thus, it is important that researchers using these reagents present as much information as possible so their results are not dismissed by skeptics in the field. Some suggestions include: showing by Western blot that the Sox2 and pax7 MOs significantly decrease protein expression perhaps in a cell line in which the increased amount of MO would not be lethal as it is in the embryo; showing that the MOs will not bind to closely related family members, i.e., simply do a sequence comparison; rewording the legend of Figure 5 to state what percent KD is achieved. Yes, the bar graphs show those data, but the text could be misconstrued. For example, on first read I misinterpreted the figure legend to state that Pax7MOs achieved 81% knock-down. I think it would be beneficial to make the language a little clearer.*

To address this point, we now clarify in the methods section that the UTR sequences of Sox1, Sox2 and Sox3 and very different, as are the sequences of Pax3 and Pax7. To further demonstrate specificity and efficacy, we have added additional controls for the morpholinos, now shown in a new supplementary figure. Specifically, we show that the Sox2 MO causes knock-down of a construct with the Sox2 UTR sequence driving RFP expression but has no effect on an analogous construct with the Sox3 UTR. Similarly, a Pax7 MO affects expression driven by Pax7 UTR but has little effect on a Pax3 UTR. Moreover, the Pax7 morpholino has been previously used and validated (Basch et al., 2006; Simoes-Costa et al., 2012). We have rewritten the results to be more clear and thank the reviewers for pointing out the lack of clarity.

Importantly, while doing these experiments, we noticed sequence polymorphism in the Pax7 UTR with an extra “C” within the morpholino recognition site. Therefore, we tested the knock-down ability of both the published UTR sequence and the one with the extra C. In both cases, we observed knock-down though the former was more robust.

We thank the reviewer for pointing out the confusing language when describing results of the knock-downs and we have now reworded the text and figure legends to be more clear, as well as included data from these additional important controls.

*6) It has been well described in the Drosophila gap gene literature that mutually repressive TFs are expressed in the same cells and regulation of their relative levels eventually leads to sharp segmental boundaries. I see a lot of similarities between that work and what is presented in this manuscript, particularly since Sox2 and Pax7 seem to repress each other. Therefore, reference to this mechanism is appropriate.*

This is an excellent suggestion and we have now incorporated reference to this type of mutual repression in the Discussion.

*7) The number of embryos is rather on the low side; for analytic power in mouse embryos we use 5 embryos per experimental group/genotype as the rule of thumb. Since chick embryos are more readily available, the conclusions would be more convincing if the power were increased by larger sample sizes.*

We have now indicated numbers of embryos and have increased the sample size.